# Alternative splicing controls teneurin-3 compact dimer formation for neuronal recognition

Christos Gogou [1], J. Wouter Beugelink [2], Cátia P. Frias [1], Leanid Kresik [1], Natalia Jaroszynska [3], Uwe Drescher [3,4], Bert J. C. Janssen [2], Robert Hindges [3,4] & Dimphna H. Meijer [1] ✉

Neuronal network formation is facilitated by recognition between synaptic cell adhesion molecules at the cell surface. Alternative splicing of cell adhesion molecules provides additional specificity in forming neuronal connections. For the teneurin family of cell adhesion molecules, alternative splicing of the EGF-repeats and NHL domain controls synaptic protein-protein interactions. Here we present cryo-EM structures of the compact dimeric ectodomain of two teneurin-3 isoforms that harbour the splice insert in the EGF-repeats. This dimer is stabilised by an EGF8-ABD contact between subunits. Cryo-EM reconstructions of all four splice variants, together with SAXS and negative stain EM, reveal compacted dimers for each, with variant-specific dimeric arrangements. This results in specific *trans*-cellular interactions, as tested in cell clustering and stripe assays. The compact conformations provide a structural basis for teneurin homo- and heterophilic interactions. Altogether, our findings demonstrate how alternative splicing results in rearrangements of the dimeric subunits, influencing neuronal recognition and likely circuit wiring.

During the formation of the central nervous system, specific guidance and recognition signals are required to orchestrate the billions of outgrowing neurons as they form complex circuits. Important cues are especially provided by combinations of cell adhesion molecules expressed on the surface of neurons. Among these, teneurins, a conserved family of type II transmembrane proteins, play a crucial role in circuit wiring across various species[1]. However, the molecular mechanisms of how teneurins facilitate the formation of functional neuronal circuits remain poorly understood.

In mammals, there are four family members of teneurins, each expressed in distinct and often interconnected areas of the developing nervous system[2,3]. They have the capability to form homophilic interactions with other teneurins[4-8], as well as heterophilic interactions with other synaptic proteins, enabling a diverse range of functions in circuit development and maintenance[9]. For instance, the interaction between mouse teneurin-3 (Ten3) and latrophilin-2 (Lphn2), an adhesion G-protein coupled receptor (GPCR), orchestrates proper circuit wiring in the hippocampus[7]. This interaction involves a Ten3-Lphn2 repulsive mechanism, where axons expressing Ten3 avoid connecting with Lphn2-expressing regions. Instead, Ten3-expressing axons grow toward other areas enriched in Ten3, pointing toward homophilic *trans*-synaptic interactions[7,10]. Additional players such as FLRT proteins and Glypican-3 may contribute to ternary complex formation with teneurins to control diverse functions such as neuronal cell migration, synapse formation, and pruning[6,11,12]. Finally, genetic abnormalities of all members of the

[1]Department of Bionanoscience, Kavli Institute of Nanoscience Delft, Delft University of Technology, van der Maasweg 9, Delft, the Netherlands. [2]Structural Biochemistry, Bijvoet Centre for Biomolecular Research, Faculty of Science, Utrecht University, Universiteitsweg 99, Utrecht, the Netherlands. [3]Centre for Developmental Neurobiology, King's College London, Guy's Campus, London, UK. [4]MRC Centre for Neurodevelopmental Disorders, King's College London, London, UK. ✉e-mail: d.h.m.meijer@tudelft.nl

teneurin family have been associated with diverse neurological disorders[13]. For example, teneurin-1 is linked to autism[14], teneurin-2 to major depressive disorder[15], teneurin-3 to microphthalmia[16], and teneurin-4 to essential tremor[17] and schizophrenia[18].

The teneurin intracellular domain (ICD), approximately 45 kDa in size, exhibits less conservation across different homologues compared to the extracellular domain (ECD)[19]. However, common features and domains can be identified, such as an EF-hand-like $Ca^{2+}$-binding site, potential phosphorylation sites, and polyproline-rich regions known to bind SH3-containing adaptor proteins[20]. The ICD may be released by a mechanism known as regulated intramembrane proteolysis and result in translocation of the ICD to the nucleus where it can coordinate transcription[20–22]. Structural studies of the teneurin ectodomains have revealed a multi-domain ~1850-residue core superfold, prominently featuring a tyrosine-aspartate rich barrel-shaped structure termed YD-shell[23,24]. This YD-shell is adorned with a calcium-binding C-rich domain, a transthyretin-like (TTR) domain and a fibronectin-like region (FN-plug), as well as with a 6-bladed β-propeller NHL domain (NCL-1, HT2A, Lin-41), all located at its N-terminal end[23,24]. At the C-terminal side of the ectodomain, structurally adjacent to the YD-shell, lies an antibiotic-binding domain-like (ABD) fold and a domain akin to the Tox-GHH class of DNases[25]. Connecting the transmembrane helix with the core superfold are a predicted immunoglobulin (Ig)fold, and eight predicted EGF domains (EGF1-8). Notably, EGF2 and EGF5 contain free cysteines that directly contribute to covalent *cis*-dimers formation in all teneurins[26,27]. A homophilic compact *cis*-dimer was resolved for human teneurin-4, which demonstrated a key role for the ABD domain in the dimer interface[5]. In this compact conformation, the ABD domain contacts the C-rich domain, the YD-shell, and the ABD domain of the other subunit (subunit refers to one of the two chains in the covalent homodimer). Further X-ray crystallography analysis of this teneurin-4 C-rich domain has revealed that three calcium ions are coordinated by eight acidic residues in the C-rich domain. These residues are conserved in mouse and human Teneurin paralogues. SAXS analysis showed that calcium binding stabilizes the compact dimer conformation of teneurin-4. More recently, Li and coworkers reported a *Drosophila* teneurin (Ten-m) homodimer connected through a TTR-NHL interface[28]. This asymmetric complex was shown to be capable of forming zipper-like higher-order oligomers, most probably as linear superstructures. Two subsequent studies detailed the structural basis of the teneurin-latrophilin interaction. Both works revealed a binding site for the Lec domain of latrophilin on the YD-shell of teneurin-2, directly opposite to the ABD and Tox-GHH domains[11,29]. Together, these structural studies unveiled the intricate architecture of teneurin ectodomains, showcasing their multi-domain core superfold and revealing critical insights into their dimeric conformations and interactions, both homo- and heterophilic.

Often, alternative splicing of CAMs plays a crucial role in diversifying their functions and fine-tune cellular interactions. For instance, several synaptic CAMs, such as Dscam, the LAR-RPTP family, neurexin, and neuroligin, harbour few-residue splicing variations that change *trans*-cellular binding sites directly[30–39]. The ectodomains of teneurin proteins also contain two alternatively spliced regions that may combinatorially contribute to the code for molecular recognition at the synapse. The Ten3 splice inserts reside between EGF7 and EGF8 (AHYLDKIVK in mouse Ten3), referred to as splice insert A, and between the first and the second blade of the NHL β-propeller (RNKDFRH in mouse Ten3) referred to as splice insert B. Berns and colleagues performed cDNA sequencing of postnatal mouse brain to demonstrate that all teneurin-3 A and B splice variants are expressed in mouse subiculum, and all except $A_0B_0$ are found in the CA1 region of the hippocampus[10]. Functionally, the alternative splice variants of teneurin-3 were found to affect binding partner specificity[6,24,29]. Namely, mouse Ten3-$A_0B_0$, lacking both splice inserts, is not capable of homophilic cell clustering in K562 cells, whereas the presence of either

or both splice inserts ($A_1B_0$, $A_0B_1$, $A_1B_1$) was reported to induce clustering in homophilic fashion[10]. Interestingly, Ten3-$A_0B_0$ (as well as Ten3-$A_1B_1$, other variants not tested) is capable of inducing heterophilic clusters with latrophilin-3[10]. Alternative splicing of teneurin-2 at similar sites has also been functionally characterised. Teneurin-2 without splice insert B (Ten2-$A_0B_0$) was shown to induce cluster formation with latrophilin-3-expressing HEK cells, whereas the presence of splice insert B in teneurin-2 (Ten2-$A_0B_1$) obstructed cluster formation with latrophilin-3-expressing cells[24,29] Strikingly, this difference in latrophilin binding was only observed with full-length, membrane-tethered, teneurin-2 but not when teneurin-latrophilin binding was assessed using soluble ectodomain variants. This suggests a shape-shifting mechanism in which the latrophilin binding site becomes inaccessible in the presence of splice insert B due to conformational restraints, rather than slice insert B changing the latrophilin binding site directly. In addition, HEK-neuron co-culturing experiments revealed that overexpression of Ten2-$B_0$ induces excitatory post-synaptic specialisations in neurons, whereas overexpression of Ten2-$A_0B_1$ induces inhibitory synapse formation[29].

It is therefore evident that the role of alternative splicing in teneurins is crucial for controlling *trans*-cellular interactions, yet the structural basis of splicing-dependent assembly formation remains unclear. Here, we present a structural comparison of four splice variants of mouse teneurin-3 full ectodomains. Using single particle cryo-electron microscopy (cryo-EM), in combination with small-angle X-ray scattering (SAXS) and negative stain EM, we reveal a homo-dimeric interface for teneurin-3 $A_1B_0$ and $A_1B_1$ ($A_1$ variants), resulting in a compact dimer conformation. This interface is stabilised by an extended β-sheet interaction between EGF8 and the ABD domain of interacting subunits. Also, the $A_0B_0$ and $A_0B_1$ variants ($A_0$ variants) are compact dimers in solution but they are distinct from the compact conformation for the $A_1$ isoforms. For the $A_0$ variants, EGF6 and EGF7 are additionally present in the reconstructed cryo-EM density map, whereas for $A_0B_0$ a contact site is observed between the EGF6 and the YD/ABD domains. Notably, each $A_0$ dimer displays a different arrangement of the two core superfolds. Finally, cell clustering and neuronal stripe assays highlight the capability of the A and B splice insert to directly mitigate *trans*-cellular interactions in vitro. These findings support a model wherein the subunits of the Ten3 dimers adopt different orientations due to the presence of relatively small splice inserts. These structural rearrangements, in turn, expose different binding interfaces that play a crucial role in orchestrating cellular interactions essential for specifying neuronal circuitry formation.

## Results

### $A_1$ variants compact dimerise through an extended β-sheet

To examine the structural basis of the alternative splice variant $A_1B_1$ of teneurin-3 (Ten3-$A_1B_1$), we expressed the complete extracellular domain (ECD, residues 342-2715) of N-terminal hexahistidine-tagged mouse Ten3-$A_1B_1$ in HEK293-E cells (Fig. 1A). We purified the proteins by Ni-NTA affinity chromatography followed by size-exclusion chromatography in the presence of calcium (Supplementary Fig. 1A). The purified ECD forms a constitutive dimer, linked by two disulphide bonds in EGF2 and -5 (Cys549 and Cys648, respectively, Fig. 1A). The dimer structure was resolved to 3.1 Å using single particle cryo-electron microscopy, revealing a compact conformation (Fig. 1B, Supplementary Fig. 1B–D, and Supplementary Data 1). The core of the compact dimer could be further resolved to 3.0 Å (Supplementary Fig. 1D). The ECD (residues 342-2708, Fig. 1A) of the $A_1B_0$ isoform was additionally purified and reconstructed by cryo-EM, resulting in a density map with 3.9 Å overall resolution (Supplementary Fig. 2). In the compact conformation, the NHL domains are facing away from each other (Fig. 1B and Supplementary Fig. 2E, middle) and the two subunits are in-plane when viewed from the NHL domain (Fig. 1B and Supplementary Fig. 2E, right). The total dimensions of the Ten3-$A_1B_1$ compact

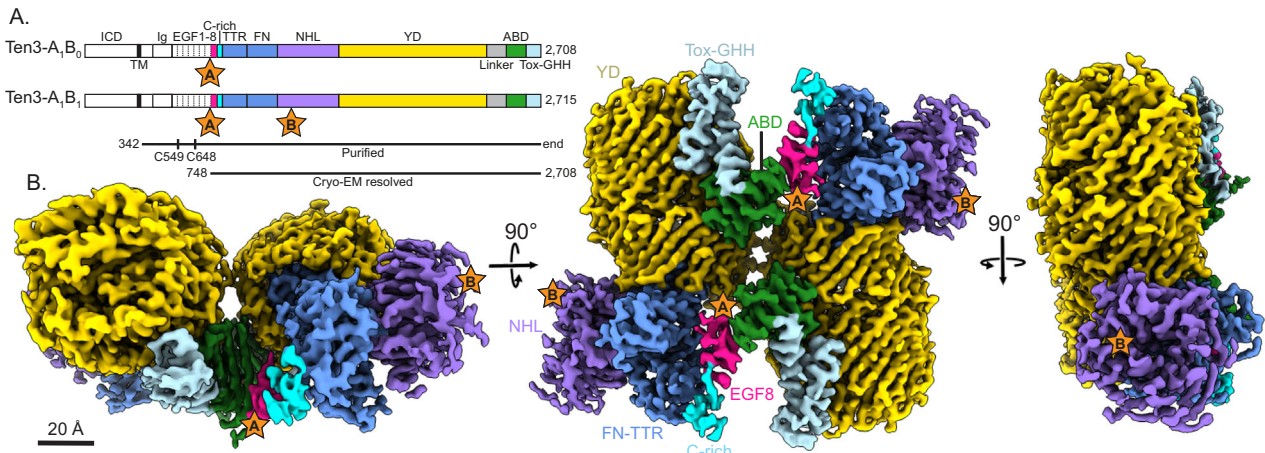

**Fig. 1 | Teneurin-3 $A_1B_1$ core superfolds form a compact dimer. A** Linear representation of the full-length teneurin-3 $A_1B_0$ and $A_1B_1$ domain compositions. Splice inserts are indicated as stars: splice insert (**A**) in between EGF7 and EGF8; splice inserts (**B**) in the NHL domain. The expressed and purified ectodomain for both isoforms are indicated with the black line below. Free cystines in EGF2 (C549) and EGF5 (C648) that enable constitutive dimerisation are indicated. The bottom black line indicates domains (EGF8 through Tox-GHH) resolved for the $A_1B_1$ isoform using cryo-EM. ICD, intracellular domain; Ig, immunoglobulin fold; EGF, epidermal growth factor repeat domains 1 through 8; C-rich, cysteine-rich region; TTR, transthyretin-related; FN, fibronectin plug; NHL, NCL, HT2A and Lin-41; YD, tyrosine-aspartate; ABD, antibiotic-binding domain; Tox-GHH, toxin-glycine-histidine-histidine. **B** Three different views of the density map of the teneurin-3 $A_1B_1$ compact dimer at 3.1 Å resolution (EMD-18889). Domain annotations are coloured corresponding to the linear representation in A). Positions of the splice inserts are indicated with stars.

in-plane dimer are 122 Å by 141 Å by 70 Å. The interface is mainly composed of hydrophilic and hydrophobic surfaces (Supplementary Fig. 4A).

An atomic model was built into the 3.1 Å resolution $A_1B_1$ dimer map. The main compact dimer interface is formed through an extended edge-to-edge β-sheet between residues 761-764 in EGF8 and residues 2594-2599 in the ABD of the two respective subunits (interface I, Fig. 2A–E and Supplementary Fig. 3A). This interface is only permitted upon displacement of a β-strand on the ABD side, which remains present in a non-compacted subunit reconstruction (Fig. 2C, and Supplementary Fig. 4B–D). Interestingly, a mutation associated with microphthalmia in humans, namely Arg2563Trp in human Ten3[16] – corresponding to Arg2579 in mouse Ten3 -, is located directly at the border of the missing β-strand (Fig. 2C). To further stabilise the extended β-sheet, Ser758-Thr2600 side chains undergo hydrogen bonding, and Arg761 forms a salt bridge with Asp2611 and a hydrogen bond with Ser2598 (Fig. 2D). The EGF8-ABD interface is also supported by hydrophobic interactions between Trp769 and Leu764 on EGF8, and Leu2575, Leu2578, and Val2595 on ABD (Fig. 2D–E). Splice insert A residues Val748-Lys749 are resolved at the N-terminal side of EGF8 (Fig. 2D), and focussed classification of all dimer particles results in extra connected density that folds back onto the ABD domain, which presumably represents the remainder of the insert (Supplementary Fig. 1D). The compact dimer is further stabilised by YD-ABD (interface II, Fig. 2A and F, and Supplementary Fig. 3B) and YD-YD (interface III, Fig. 2A, B, G, and Supplementary Fig. 3C) contacts. Interface II contains Thr1607-Asn2589 and Lys1612-Asn2534 hydrogen bonding (Fig. 2F). Interface III contains a Thr1604-Gly1618 hydrogen bond (Fig. 2G). Individually, the interface-contributing residues in EGF8 span a total interface area (IA) of 420 Å², those in the YD-shell 476 Å², and 657 Å² in the ABD domain (Fig. 2B). A separate subset of dimer particles harbours no YD-YD interactions, resulting in relatively flexible YD domains hinging at the C-rich/EGF8 region ("open conformation", Supplementary Fig. 1D, Supplementary Movies 1 and 2). The $A_1B_0$ (Supplementary Fig. 2E) compact dimer is found in this open conformation – lacking interface III – as opposed to the $A_1B_1$ closed compact dimer (Fig. 1B) that includes this additional YD-YD contact.

What are the differences between the mouse teneurin-3 compact dimer presented here and the human teneurin-4 (Ten4) compact

dimer published previously[5]? The main difference between the Ten3 and Ten4 structures is the orientation of the two subunits with respect to each other (Fig. 3A). The subunits in the human Ten4-$A_1B_1$ dimer are each turned outwards at a 30° angle, thereby creating a 'transverse dimer' (Fig. 3A). The subunits in the compact dimer configuration of mouse Ten3-$A_1B_1$, however, align in the same plane (Fig. 1B). In both structures, the ABD domain is central in the interface, but different interactions are formed. In the human Ten4 transverse dimer, the ABD domain contacts the C-rich domain, the YD shell and the ABD domain of the other subunit. In the dimer of mouse Ten3, the ABD domain contacts EGF8 and the YD shell (Fig. 3B). The difference in dimer conformation results in a molecule with smaller dimensions for mouse Ten3 compared to human Ten4 (122 Å by 141 Å by 70 Å versus 133 Å by 166 Å by 112 Å, respectively). A detailed structural comparison reveals that similar residues on the ABD domain and the YD shells are involved in both the transverse versus and in-plane dimer interactions (Fig. 3C). Interestingly, a centrally positioned ABD loop in Ten4 interface – could not be resolved in the in-plane Ten3 compact dimer structure (missing Asn2531-Ala2533, Fig. 2F). Altogether, our cryo-EM density map reveals a compact conformation of covalently dimerised mouse Ten3, which is different from previously published dimeric conformations of teneurin homologues[5,23,28].

## Splice insert A as a potential spacer to dislodge EGF6

To investigate the effect of splice insert A on the architecture of the EGF-repeat domains, we expressed and purified the full ectodomains and acquired cryo-EM data of all splice variant combinations of mouse Ten3: Ten3-$A_0B_0$, $A_0B_1$, $A_1B_0$ and $A_1B_1$ (Fig. 4A, Supplementary Fig. 5A-C, and Supplementary Data 1). Full ectodomain structures of compact dimers could not be resolved for the $A_0$ variants, presumable due to air-water interface issues during cryo-EM grid preparation. Similar issues were also observed previously for the Ten4 dimer[5]. Instead, we resolved non-compacted subunits containing partial EGF-repeat domains for each of the proteins (Fig. 4B and Supplementary Fig. 5D). Comparison of the four structures reveals that the subunits adopt similar conformations with the largest differences in the orientation of EGF6 and EGF7. In absence of splice insert A ($A_0B_0$ and $A_0B_1$ isoforms), EGF-repeats 6-8 run across the FN domain towards the YD shell, whereby EGF6 is located in a groove

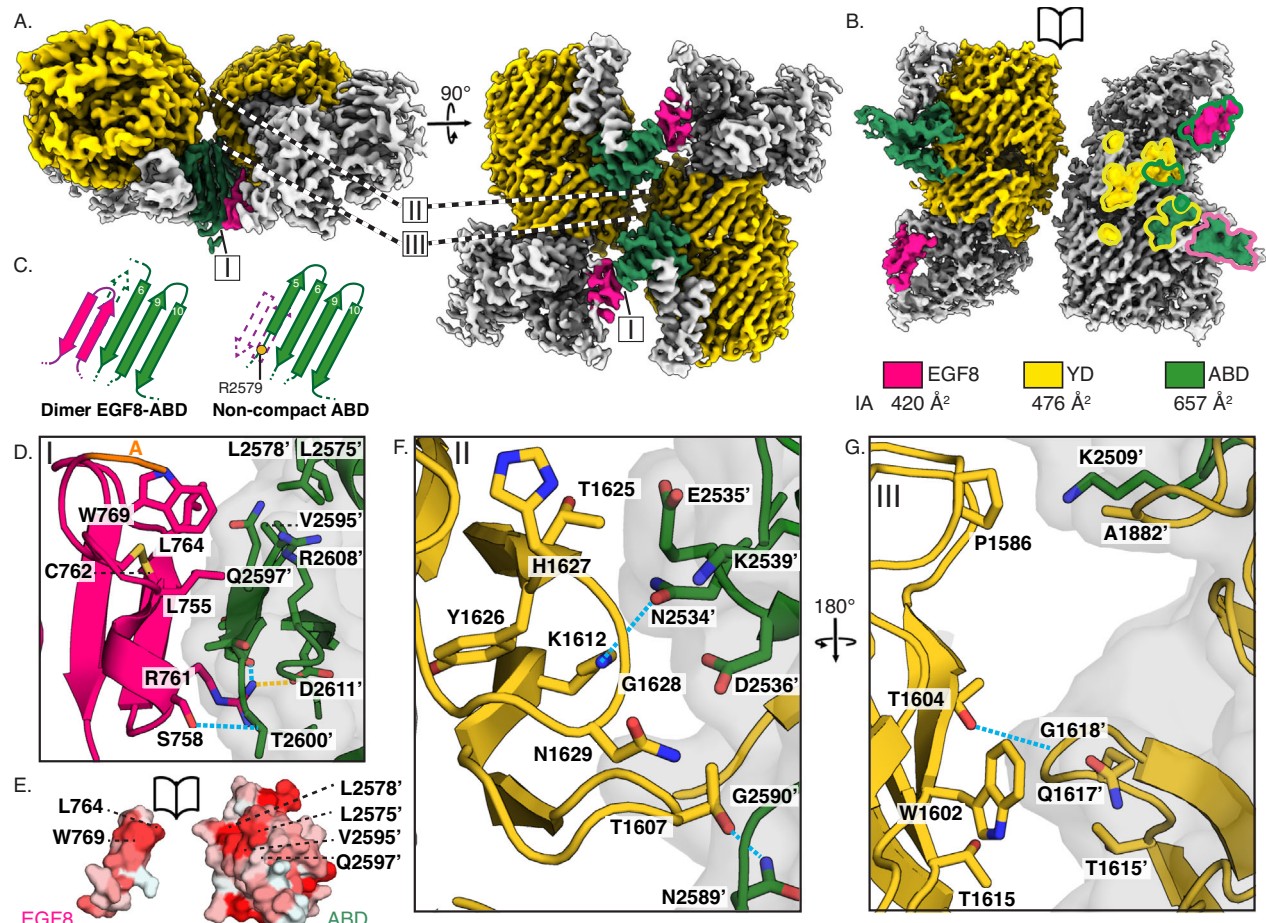

**Fig. 2 | The teneurin-3 $A_1B_1$ compact dimer interface comprises three distinct interaction sites. A** Density map of the Ten3-$A_1B_1$ compact dimer with domains that participate in the interfaces shown in colour. The colour code is shown below panel **B**, along with the interface area (IA) per domain. **B** Open book representation of the density map. The left subunit shows the participating domains in colour, and the right subunit displays the specific interface residues. Fill colours indicate the domains in which the residues are located, and outline colours indicate the domains in which the contacting residues reside. **C** Schematic representation of EGF8-ABD interface in the compact dimer versus the ABD domain of a Ten3 $A_1B_1$ non-compact subunit reconstruction. Residues 2579-2588 are displaced or disordered in the compact dimer form, and residues 2582-2587 form β-strand 5 in ABD of the non-compact subunit. **D** Close-up of interface I, the EGF8-ABD extended β-sheet (PDB:8R50). Resolved residues from insert A (labelled A) are coloured orange **E** Open book surface hydrophobicity representation of the EGF8-ABD interface. **F** Close-up of interface II, comprising YD-ABD contacts. **G** Close-up of interface III, the YD-YD and YD-ABD contacts. In the close-ups, ionic bonds are indicated in dashed yellow lines and hydrogen bonds in dashed blue lines, and transparent surface representation is used to distinguish between subunits within the compact dimer. Only residues with >5% per-residue buried surface area are represented. For clarity, Leu2530, Ser2596, and Arg 2624 are not displayed.

between the YD shell and ABD domain of the same subunit (Fig. 4C). EGF6 directly contacts several negatively charged residues on the YD β-barrel and Gln2506 in the ABD domain. However, in the presence of splice insert A ($A_1B_0$ and $A_1B_1$ isoforms) – predicted to be a 9-residue helix inserted in the linker between EGF7-8 -, EGF6 and EGF7 cannot be resolved anymore, suggesting increased flexibility of EGF6 with respect to the YD shell. Possibly, this splice insert acts as a physical spacer to dislodge EGF6 out of its binding groove, thereby preventing the EGF6-YD/ABD interface from forming. Low-resolution EM density of the potential splice insert A can indeed be observed hovering over the ABD domain (Supplementary Fig. 1D), thus dislodging EGF6 from its YD/ABD groove. The increased rigidity of EGF6 and EGF7 with respect to the YD shell in the absence of splice insert A appears to hinder EGF8 from interacting with the ABD domain on the other subunit (Fig. 4D, E) –an interaction necessary for the compact dimerisation observed in $A_1B_1$ (Fig. 2C–E). Thus, depending on splice insert A, the EGF-repeat region reorganises relative to the core superfolds, with potential implications for the orientation of the two subunits themselves and further homo- or heterodimeric interactions.

## All isoforms are compact in a calcium-dependent manner

To further investigate the splicing-dependent effects on the conformation of the Ten3 dimers in solution, we performed small-angle X-ray scattering (SAXS, Fig. 5A, B and Supplementary Fig. 6A–D). In presence of calcium, all isoforms are compact in solution (Fig. 5A, B, and Supplementary Fig. 6A–D), with average radii of gyration (Rg) 7.3, 8.0, 7.7, and 7.9 nm, for Ten3-$A_0B_0$, $A_0B_1$, $A_1B_0$, and $A_1B_1$, respectively (Fig. 5A). This relation in compactness is additionally reflected in the $D_{max}$ values per isoform (Fig. 5B). The absence of calcium, as well as the addition of calcium chelator EDTA, result in increased Rg and $D_{max}$ for all isoforms, as well as a loss of the distinct compact-protein profile in the dimensionless Kratky plots and pair-distance distributions (Supplementary Data 2, Supplementary Fig. 6A-D). These effects are similar to the transition of Ten4 from a compact conformation into a more elongated conformation in the absence of calcium[5]. Notably, in the presence of calcium, out of the four splice variants $A_0B_0$ is the most compact, whereas $A_0B_1$ deviates most from a compact form (Fig. 5A, B). The increased compactness of $A_0B_0$ could possibly be explained by a predicted interaction between any of the flexible Ig or EGF domains and the splice site B within the NHL domain

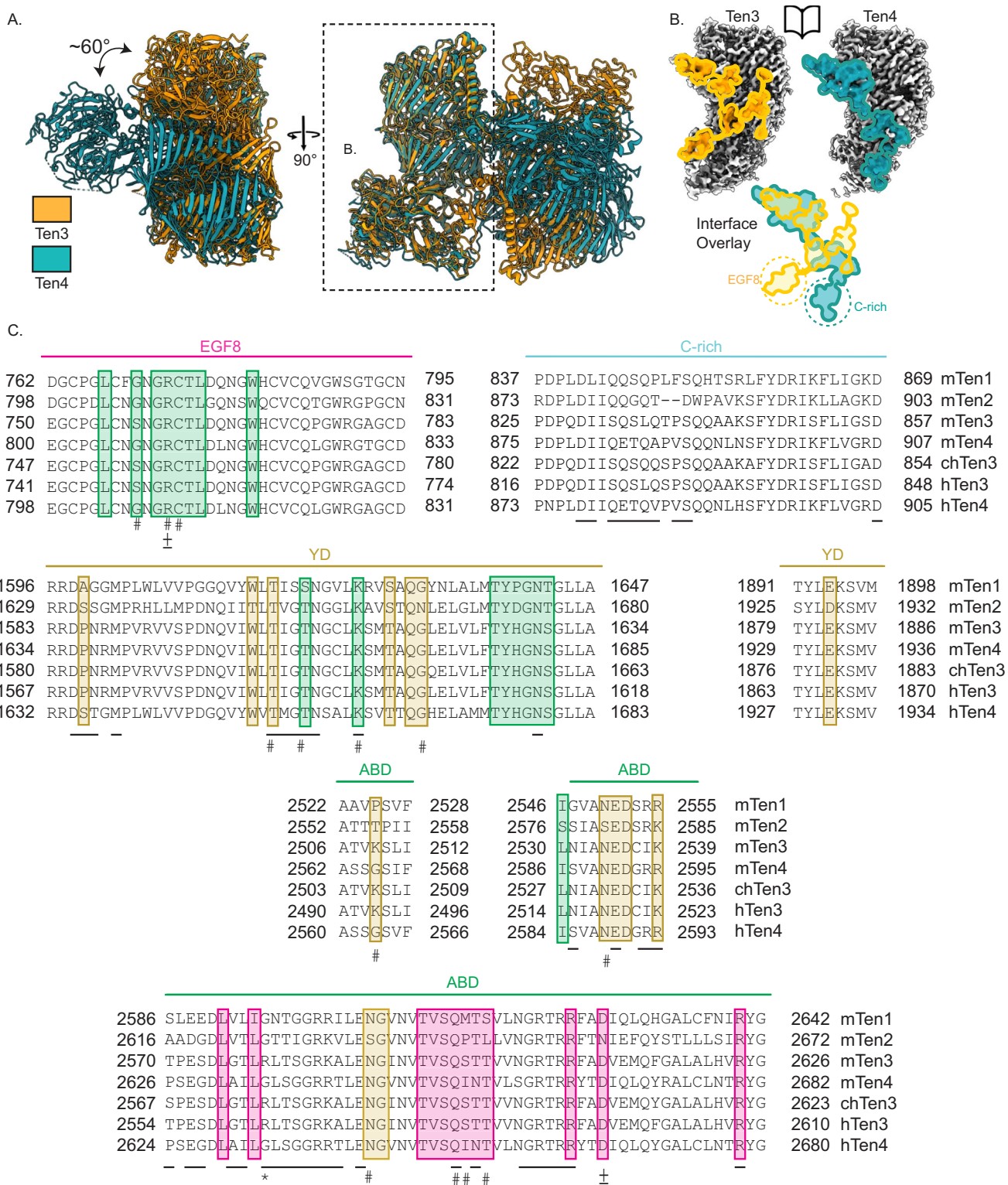

**Fig. 3 | Teneurin-3 and teneurin-4 compact dimer interfaces are different.**
**A** Alignment of one subunit from the teneurin-3 (Ten3, PDB:8R50, orange) compact dimer with one subunit from the teneurin-4 (Ten4, PDB:7BAM, blue) dimer. The remaining subunits from each dimer orient at a relative angle of approximately 60°. The dashed outline indicates the subunit represented in panel **B**. **B** Comparison of the mouse Ten3 and human Ten4 compact dimer interface projections on open-book subunits. Below are overlays of the interface projections. Dashed circles indicate the domains that exclusively participate in one or the other compact dimer. **C** Teneurin alignment across orthologues of the interface residues for the EGF8, C-rich, YD and ABD domains. Boxes around the residues indicate the contacting domains in mouse teneurin-3. Domain colouring corresponds to that in Fig. 1. Only residues with >5% per-residue buried surface area are represented. ±, ionic-bonding residues in Ten3; #, hydrogen-bonding residues in Ten3; _, interface residues in Ten4; *, Arg2579, the mouse homologue of microphthalmia-associated Arg2563 in humans.

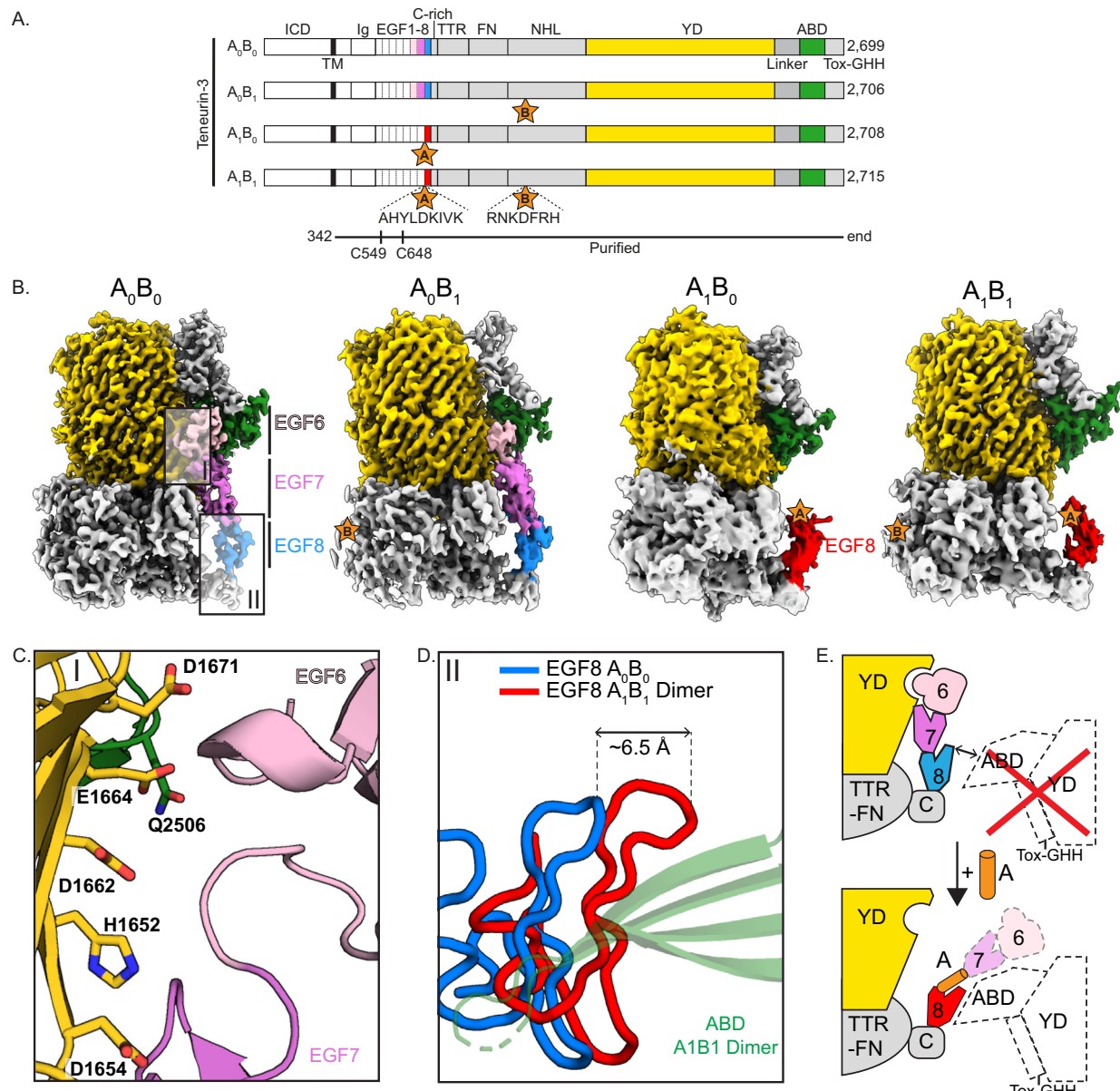

**Fig. 4 | Subunits of $A_0$ isoforms display a contact between EGF6 and the YD/ABD domains. A** Linear representations of the four full-length Ten3 isoforms with domain composition. The presence of splice inserts is indicated with stars. Domains are labelled according to Fig. 1. The expressed and purified ectodomain for each isoform is indicated with the black line below. **B** Cryo-EM reconstructions of isoform subunits (left to right: EMD-18891, EMD-18900, EMD-18902, EMD-18890). Dashed boxes I-II indicate close-ups shown in (**C**, **D**), respectively. **C** EGF6-YD/ABD interface (PDB:8R54). Interacting residues on the YD- and ABD-side of the interface are displayed and annotated. **D** Overlay of EGF8 domains of Ten3-$A_0B_0$ and $A_1B_1$. The relative displacement of EGF8 is indicated. **E** Schematic representation of dislodging by splice insert A. Top: Absence of A is accompanied by an additional interface between EGF6-YD/ABD. This stabilisation impedes EGF8 β-sheet extension with ABD for compact dimerisation. Below: Splice insert A creates distance between EGF7 and EGF8, in turn dislodging the EGF6-YD interface. Loss of this interface and the presence of splice insert A increases the flexibility of the EGF region, and EGF8 is now free to interact with the opposite ABD domain.

(Supplementary Fig. 7), which would condense the $A_0B_0$ compact dimer even further. The data further show that Ten3-$A_1B_0$ and $A_1B_1$ have highly similar SAXS profiles. Thus, it appears that splice insert B affects compact dimerisation only in the absence of splice insert A. The general compactness in the presence of calcium is supported by increased protein stability observed as a thermal shift in the protein melting curves of all splice variants (Fig. 5C and Supplementary Fig. 6E). Note that calcium-binding residues of the C-rich domain of human Ten4 are well-conserved in mouse Ten3 (Fig. 3C), suggesting that the C-rich domain of Ten3 also selectively binds calcium ions.

To corroborate the SAXS profiles indicating compact conformations for all splice variants, we collected negative stain TEM (nsTEM)

datasets for all variants. The 2D classifications of all Ten3 splice variants indeed revealed compact conformations for each variant (Fig. 5D and Supplementary Fig. 8A). Whereas Ten3-$A_1B_0$ and $A_1B_1$ have the same conformations as their cryo-EM structures (Fig. 1 and Supplementary Fig. 2E), $A_0B_0$ and $A_0B_1$ revealed alternative architectures. While the low resolution of the data precludes detailed analysis, a potential YD-YD interface is observed for $A_0B_0$ – also observed at low abundance in cryo-EM 2D classes (Supplementary Fig. 8B) -, and the $A_0B_1$ compact dimer could be explained by an FN-FN interaction. 3D reconstructions and low-resolution fitting confirmed these potential dimer interfaces (Fig. 5E). Map-model correlations were calculated for all map-model combinations, and the highest correlations were found

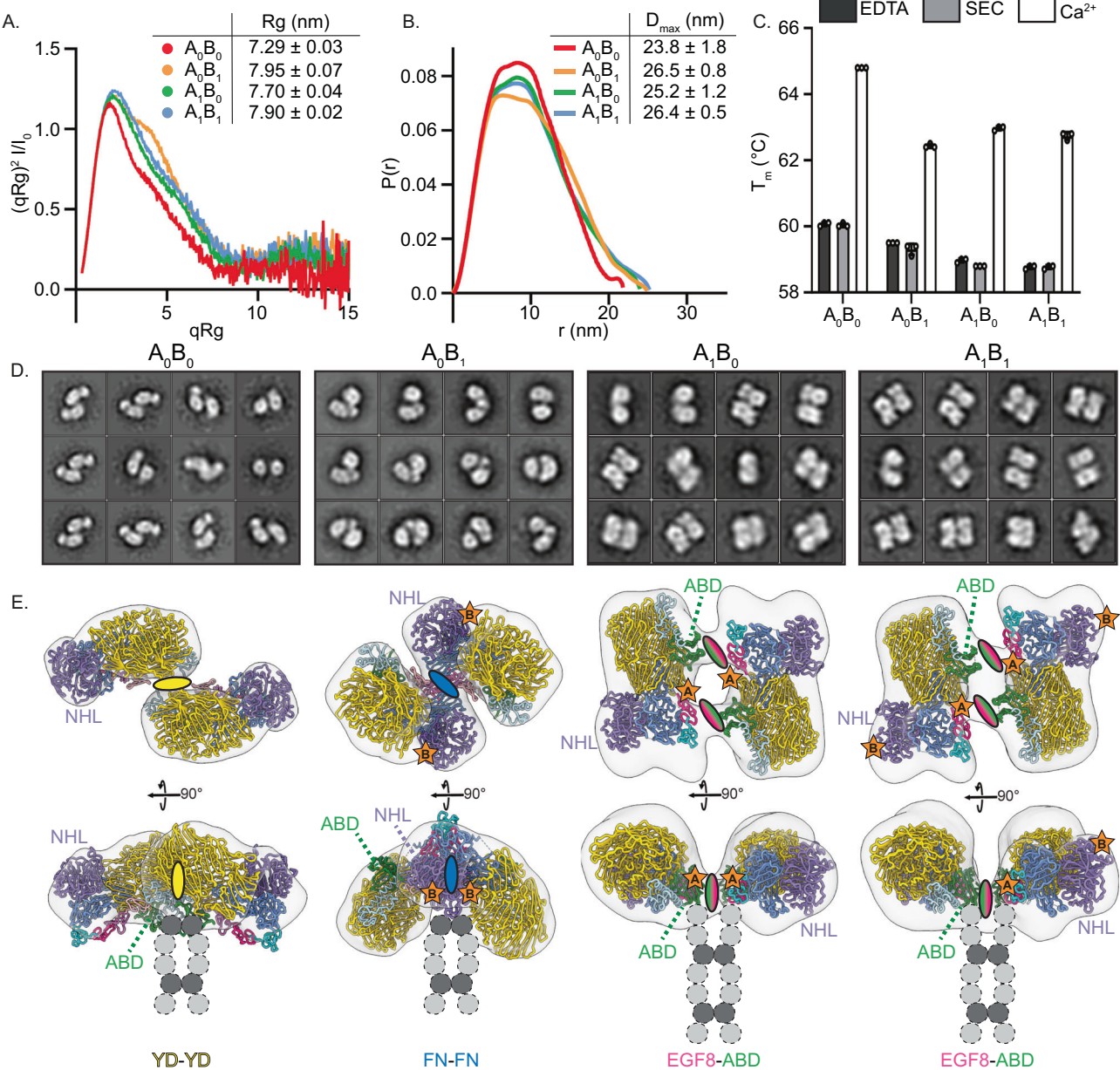

**Fig. 5 | All teneurin-3 isoforms are compact in solution with isoform-specific conformations. A** Dimensionless Kratky-plot of SAXS data for all Ten3 isoforms at 0.5 mg/mL in the presence of 2 mM $Ca^{2+}$. Inset, radii of gyration (Rg) derived from SAXS data for each Ten3 isoform averaged from three protein concentrations (mean ± SEM). **B** Pair-distance distribution for all Ten3 isoforms at 0.5 mg/mL in the presence of 2 mM $Ca^{2+}$. Inset, maximum diameter ($D_{max}$) derived from the pair-distance distribution for each Ten3 isoform averaged from three protein concentrations (mean ± SEM). **C** Melting temperatures ($T_m$) for each isoform in the presence ($Ca^{2+}$) and absence (SEC) of 2 mM calcium, as well as under addition of

5 mM EDTA (EDTA). Data are presented as mean ± SD; n=3 melting temperatures per N=1 independent experiment. Source data for (**A-C**) is provided as a Source Data file. **D** 2D classes of negative stain TEM for all Ten3 isoforms in the presence of calcium. **E** 3D reconstructions for all splice variants, shown at two different angles. Domains are coloured according to colouring in Fig. 1. Ovals indicate the location of the compact dimer interface, and the colour denotes the involved domain(s). Attachment of the EGF repeats that are not included in the map or model are added schematically as dashed circles. Disulphide-linked EGFs 2 and 5 are shown in dark grey.

between model-derived densities and the corresponding experimental densities into which the model was built (Supplementary Fig. 8C). In addition, the model for $A_1B_1$ correlates well with the map of $A_1B_0$ and vice versa, which is in line with their similarities in cryo-EM and SAXS analysis. Splice site B is not directly involved in any of the observed compact dimer interfaces, hinting at an unresolved allosteric mechanism that could explain the shifts between $A_0B_0$ and $A_0B_1$ conformations. Notably, an additional dimer interface was observed in the negative stain TEM dataset of the $A_0B_0$ variant, in which the NHL domains are connected (Supplementary Fig. 8D). This conformation

would, however, not be compatible with a compact dimer due to restraints imposed by the EGF2-2/5-5 disulphide bonds and may represent an additional higher-order *cis* contact.

We next used the SAXS profiles and compact dimer interfaces from nsTEM to model the structures of the full ectodomain of all the splice variants (Supplementary Fig. 8E). For SAXS modelling, the dimer interfaces obtained from nsTEM were restrained, whereas the domains in the EGF-Ig stalk were treated as independent connected rigid bodies, with EGF2-4 restrained to enforce the intermolecular disulphide bonds. All modelled splice variants, except for the $A_0B_1$ dimer, explain

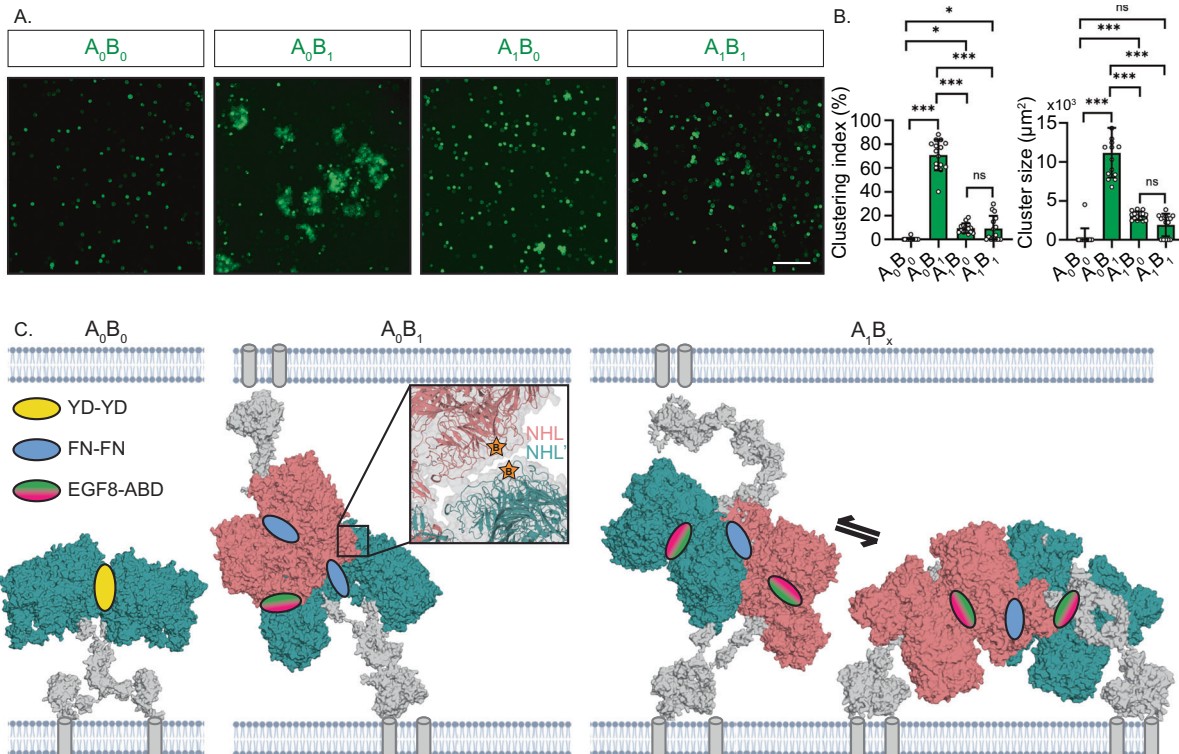

**Fig. 6 | Models for teneurin-3 homophilic *trans*-cellular complex formation.**
**A** Clustering assay of K562 hematopoietic cells electroporated with the mouse Ten3 isoforms. Scale bar is 100 µm. **B** Analysis of the clustering indices and mean cluster areas of the conditions in A. Data points for cluster size correspond to the per-image averaged values (n=5 images per N=3 independent experiments, 15 images total). Data are presented as mean ± SEM; ns: not significant, *: p < 0.05, ***: p < 0.001; P-values for clustering index comparisons from top-down are p=0.0312, p=0.0253, p<0.0001, p<0.0001, p<0.0001 and for cluster size comparisons from top-down are p=0.0983, p=0.0008, p<0.0001, p<0.0001, p<0.0001. One-Way ANOVA, Tukey's multiple comparisons test. Source data for panel **B** are provided as a Source Data file. **C)** A non-clustering Ten3-$A_0B_0$ compact dimer, a *trans*-cellular dimer-of-dimer of Ten3-$A_0B_1$, and a *trans*-cellular dimer-of-dimer of Ten3-$A_1B_x$. The $A_0B_x$ dimer-of-dimers is also compatible with a *cis* configuration due to the EGF organisation. Close-up of the $A_0B_1$ *trans* dimer-of-dimers display an additional contact between the NHL domains of subunits not involved in the EGF8-ABD contact. Dimers located on opposing membranes are distinctly coloured salmon and dark teal. The rigid-body models of each of the EGF-Ig stalks were calculated using the SAXS data of each isoform (Fig. 5 and Supplementary Fig. 8E). Trans-membrane helices are indicated by grey tubes.

their SAXS profiles well (Supplementary Fig. 8E). Possibly, the SAXS data of $A_0B_1$ cannot be explained by the nsTEM-based model, because $A_0B_1$ in solution does not exhibit the same level of compactness or adopts multiple conformations. Nonetheless, the combination of the nsTEM and SAXS data indicates at least three different dimer conformations, with $A_1B_0$ and $A_1B_1$ sharing the same EGF8-ABD mediated dimer, also observed in their cryo-EM structures, and splice variants $A_0B_0$ and $A_0B_1$ each adopting a distinct dimer conformation.

### A model for splice-dependent *trans*-interactions

The alternative conformations as shown in Fig. 5 may affect the capabilities of Ten3 splice variants for *trans*-cellular interactions. To test this, we electroporated GFP-tagged full-length transmembrane variants of all four Ten3 splice variants in K562 hematopoietic cells. *Trans*-cellular interactions were quantified by assessing cluster formation of the Ten3 electroporated K562 cells (Fig. 6A, B and Supplementary Fig. 9A). Similar to previously shown[10], all isoforms except $A_0B_0$ form homophilic clusters. In addition, we observe a quantitative difference between clustering of Ten3-$A_0B_1$, $A_1B_0$ and $A_1B_1$, whereby $A_1B_0$ and $A_1B_1$ have only limited clustering capabilities, while electroporation with $A_0B_1$ results in the highest clustering index and cluster size averages (Fig. 6B). As such, similar to the structural data (Fig. 5), splice insert B has a large effect, but exclusively when splice insert A is absent.

No *trans*-interaction is observed in the cell clustering assay for Ten3-$A_0B_0$ (Fig. 6C). Possibly, the $A_0B_0$ may not engage in *trans*-cellular interactions because it forms homophilic complexes in *cis*,

through the above-mentioned NHL-NHL contact (Supplementary Fig. 8D). To model higher-order interactions that may explain the observed *trans*-cellular interactions for the other variants, we analysed dimer-of-dimer formation of the SAXS- and nsTEM-based full ectodomain models. For compact dimer Ten3-$A_0B_1$, harbouring a potential FN-FN interface, the ABD domain would be accessible to form a *trans* dimer-of-dimer interaction with the EGF8 domain of a dimer at the opposing membrane (Fig. 6C). This interface could not be employed for dimer of dimers in *cis* due to increased constraints induced by the EGF6-ABD/YD interaction. Remarkably, the $A_0B_1$ *trans* dimer-of-dimers displays an additional NHL-NHL interface between them (close-up, Fig. 6C), with splice inserts B located at this interface. This interface is reminiscent of the teneurin-2 crystal contact reported previously[23]. Compact dimers $A_1B_0$ and $A_1B_1$ could form *trans*-cellular interactions by means of the FN-FN interface (Fig. 6C). This same interface could also be employed for an additional *cis*-interaction (Fig. 6C). A presumed equilibrium between *cis*- and *trans* dimers of dimers would explain the limited cell-clustering capability of the $A_1B_0$ and $A_1B_1$ dimers, as *cis* dimers-of-dimers would harbour no potential for *trans*-interactions. The distinct conformations of compact dimers may also have implications for their ability to bind *trans*-cellularly with latrophilin[11,29], and with FLRT in *cis*[40]. As illustrated in Supplementary Fig. 9B, $A_0B_1$ may not readily engage in *trans*-interaction with latrophilin, given that the latrophilin-interaction surface appears to be oriented towards the membrane, obstructing access to the Olfactomedin-Lectin domain. The other three splice variants seem to

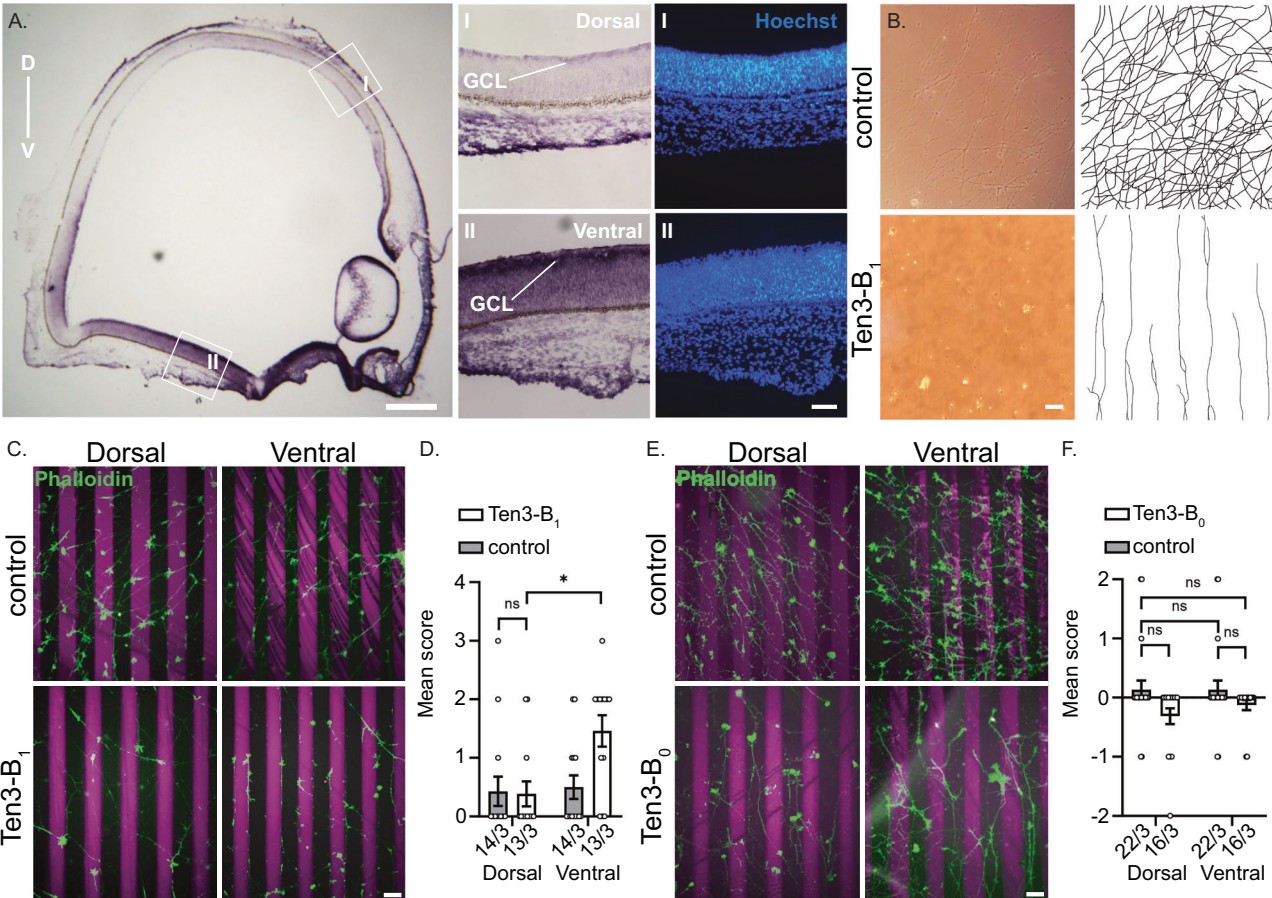

**Fig. 7 | Splice inserts B directly guides axons. A** In situ hybridisation on E6 chick retinal cryosections showing chick Ten3 mRNA expression in a gradient along dorso-ventral (*D-V*) axis. GCL, ganglion cell layer. Scale bar is 400 μm. I. Close up of dorsal (D) region of the retina showing little to no expression of Ten3, and Hoechst staining of nuclei (blue) in D retina. II. Close up of ventral (V) region of the retina showing high expression of Ten3, and Hoechst staining of nuclei (blue) in V retina. Scale bar is 50 μm. Three retinae were imaged with similar results. **B** Representative light microscope image of V RGC axons cultured on control (top) versus Ten3-$B_1$ (bottom) stripes. Scale bar is 50 μm. Right: Tracing of axons in the light microscopy images. Statistics identical to panel **D**. **C** Top: Control stripe assays with cultured retinal explants from D and V retinas. Bottom: Ten3 stripe assays with 10 μg/mL Ten3-$B_1$ (magenta), with retinal explants from D and V retina. Stripes of panels (**C**, **E**) were labelled with goat anti-Rabbit Alexa-568 secondary antibody (magenta) and RGC axons labelled with Phalloidin-Alexa Fluor 488 (green). Scale bar is 100 μm. **D** Quantification of RGC axon guidance from V vs D retina in (**C**). Guidance decisions of panels (**D**, **F**) were scored between -3, -2, -1, 0, 1, 2 or 3, reflecting varied degrees of repulsion or attraction; data of panels **D** and **F** are presented as mean ± SEM; numbers under each bar of panel **D** and **F** represent n/N, with n scored images per N independent experiments; ns: *p*=0.9992, *: *p*=0.013. Two-Way ANOVA, Tukey's multiple comparisons test. **E** Top: Control stripe assays with cultured retinal explants from D and V retinas. Bottom: Ten3 stripe assays with 10 μg/mL Ten3-$B_0$ (magenta), with retinal explants from D and V retina. Scale bar is 100 μm. **F** Quantification of RGC axon guidance from V vs D retina in panel **E**. P-values from top-down left-right are *p*=0.5794, *p*>0.999, *p*=0.1025, *p*=0.5794; ns: not significant. Two-Way ANOVA, Tukey's multiple comparisons test. Source data for panels **D** and **F** are provided as a Source Data file.

be more conducive to latrophilin binding. This observation aligns with earlier findings that indicated the incapability of a teneurin-2 $A_0B_1$ version to bind latrophilin, and thus provides a structural basis for the proposed shapeshifting mechanism[7,10,29]. None of the membrane-tethered compact dimer conformations occlude the latrophilin and FLRT binding sites (Supplementary Fig. 9B). However, FLRT may not engage in binding $A_0B_1$ if the teneurin-latrophilin trans interaction is sterically obstructed by the intermembrane setting[40].

### Insert B attracts teneurin-3-expressing outgrowing axons

We hypothesised that the emergent NHL-NHL contact in Ten3-$A_0B_1$ (close-up Fig. 6C) could provide additional *trans*-cellular stabilisation in a B-dependent manner. To address this hypothesis and whether this contact by itself provides sufficient adhesive strength, we set up chick retinal stripe assays[41,42]. First, we characterised Ten3 expression patterns in the chick retina using RNA fluorescent in-situ hybridisation (FISH). We observed that Ten3 mRNA is highly expressed in the ventral, but not dorsal, retina. (Fig. 7A). Then, to address whether Ten3 acts as

an attractant or repellent for neurons from either retinal region, we plated retinal chick explants from both regions and grew them on Ten3 stripe assays. To directly evaluate the impact of splice insert B on *trans*-cellular interactions, we utilised a monomeric variant of mouse Ten3 lacking the Ig and EGF domains (lacking residues 1-845). This specific monomer is incapable of establishing EGF8-ABD interactions, as shown in this work, to restrict the influence of splice insert B. Furthermore, the designed Ten3 monomeric variant cannot participate in the $A_0B_1$ EGF8-ABD-based dimers-of-dimers *trans*-interaction, thereby isolating the contribution of B in direct *trans*-cellular binding. At 2-3 days after explant growth, it was observed that ventral neurons localised preferentially to the Ten3-$B_1$-covered regions with their axons growing parallel to the stripes, whereas the dorsal neurons covered the entire surface with axons growing in random directions (Fig. 7B, D). No such difference was observed when culturing ventral or dorsal neurons on Ten3-$B_0$ stripes (Fig. 7E, F). Notably, when only labelled and unlabelled control antibody was immobilised in the stripe assay, neither the ventral nor the dorsal neurons showed any preferential growth over

particular stripes. (Fig. 7C–F). The K562 and retinal explant studies together show that, while the presence of splice insert A may allow for *trans*-cellular clustering in a manner unaffected by B, the presence of splice insert B itself can support and induce robust *trans*-cellular interactions and serves as an attractant to chick retinal neurons from the ventral region.

## Discussion

Cell adhesion molecules (CAMs) play a prominent role in the process of neuronal wiring by establishing a molecular recognition code at the synapse through their combinatorial expression pattern. Teneurins are a family of synaptic CAMs that exhibit additional *trans*-synaptic specificity through alternative splicing of two sites, referred to as splice site A and splice site B. Here, we reveal a compact dimer conformation of the ectodomains of mouse Ten3-$A_1B_0$ and $A_1B_1$. The compact dimer interface is formed through an extended β-sheet interaction between the EGF8 and ABD domains on interacting subunits. Additionally, the YD-shell and ABD domain participate in further stabilisation. We resolved the cryo-EM subunits of each splice variant, and these data demonstrate an additional interaction between EGF6 and the ABD/YD domain only when splice insert A is absent. We further provide small-angle X-ray scattering (SAXS)-based evidence, complemented with negative stain EM, that all splicing variants are compact in presence of calcium. This data corroborates that $A_1B_0$ adopts the same conformation as $A_1B_1$, while $A_0B_0$ and $A_0B_1$ display different compact conformations. These large rearrangements of the two subunits have practical implications for *trans*-cellular interactions, as demonstrated in the hematopoietic cell line K562 and chick retinal neurons. Our data provide a model where relatively short splice inserts of ~7 amino acids result in large structural rearrangements, hereby impacting *trans*-cellular interactions and neural circuit wiring.

*Trans*-synaptic adhesion is commonly regulated by alternatively spliced variants, often directly affecting the *trans*-cellular contacts[10,43–46]. For instance, the neuroligin-neurexin complex is regulated by the presence of two neuroligin splice inserts (A, 30 residues; B, 8 residues) that directly occlude neurexin binding[37]. Wilson and colleagues further revealed that the presence of a neurexin splice insert (SS2A, 8 residues) elongates a β-sheet near the interface with neurexophilin, directly stabilising this complex[38]. Similarly, two so-called micro-exons A and B (meA, 9 residues; meB, 4 residues) strongly determine *trans*-synaptic binding for PTPδ, PTPσ, and LAR, three members of the LAR-RPTP family[46]. The inserts were shown to either directly remodel the *trans*-synaptic interface[30–38], or - in the case of meB as a linker between Ig2 and Ig3 - by increasing the flexibility of Ig3 relative to Ig1/Ig2[39], thereby extending the interface. These inserts then determine selective binding of PTPδ to IL1RAPL1 and IL-1RAcP[39], Slitrk1/2[34,35], and SALM5[36], and the interactions of PTPσ with SALM3[47] and TrkC[48] [a]. Similarly, cell-specific expression of Dscam splice variants – with a total of 38,016 possible Dscam isoforms - underlies dendritic self-avoidance[49], and various structural works have shown how Dscam splicing directly remodels the homophilic binding interface to orchestrate different *cis* and *trans* interactions[30–33]. These findings are all examples in which the *trans*-cellular binding interface is directly modulated by small splice variations.

In contrast, our data illustrate how alternative splicing of teneurin-3 results in large rearrangements of the individual subunits, referred to as shapeshifting by Li et al.[29], forming different compacted dimers, and thereby exposing new binding sites. Specifically, the presence of splice inserts A increases the flexibility of the EGF repeat region, thereby facilitating EGF8-ABD compact dimers to form. Additionally, the presence of splice insert B facilitates an FN-FN compact dimer, but only in the absence of splice insert A. Our data do not explain why Ten3-$A_0B_0$ and Ten3-$A_0B_1$ are structurally different. Although splice insert B locally only modifies the loop in which it is located (Supplementary Fig. 7A), we speculate that the NHL domain (including splice site B) may be engaged in direct contact with any of the Ig or EGF domains (Supplementary Fig. 7B-D). This would impose or remove additional restraints resulting in FN-FN compact dimers for the Ten3-$A_0B_1$ variant. Importantly, these different compact dimer conformations expose different surfaces available for dimer-of-dimer formations in *trans*. For instance, the EGF8-ABD compacted dimers are exposing FN domains for FN-FN *trans* interactions. Conversely, the FN-FN compact dimers are exposing the EGF8 and ABD domains for *trans* and *cis* dimers-of-dimers.

In recent years, it has become evident that the pre-and post-synaptic membranes are tightly organised[50]. For instance, post-synaptic receptors are aligned with pre-synaptic vesicle release sites, hereby forming nanodomains, and potentially enhancing the efficiency of signal transfer. Cell adhesion molecules may also play a significant role in this process by directly interacting with receptors to orchestrate the molecular organisation at the synaptic membrane. Indeed, few studies have now shown that cell adhesion molecules are also present in nanodomains, exemplified by neurexins[51,52]. More recently, using STORM super-resolution microscopy, researchers discovered that teneurin-3 is also organised in nano-size clusters of approximately 80 nm in size, localised at the pre-synaptic side[53]. It would be interesting to explore further what is required to form such 80 nm clusters, as our *cis*- and *trans*-interactions were only compatible with linear arrays. This outcome resembles what is predicted from the asymmetric dimeric contact observed in fly teneurin[28]. Presumably, the formation of such clusters may require the presence of additional cell adhesion molecules, membrane-embedded proteins, lipids, or interacting receptors.

Taken together, an underlying principle is emerging regarding the relationship between alternative splicing and neural connectivity. Teneurin's splicing-dependent conformational reorganisations themselves present a unique example of large rearrangements causing indirect remodelling of *trans*-synaptic interactions. These splicing-dependent conformation acrobatics shed light on how small changes in protein sequence realise large conformational shifts at the molecular level and orchestrate neuronal circuitry formation up to the scale of entire nervous systems. Future studies will have to address the cell-specific expression of such splice variants and the consequences for neuronal wiring specificity.

## Methods
### Ethics
All animal work was approved by King's College London Animal Welfare and Ethics Board (AWERB) under the Project licence 70/9036 to RH.

### Statistics and reproducibility
No statistical method was used to predetermine the sample size. No data were excluded from the analyses unless specifically stated for the method. The experiments were not randomized unless explicitly stated for the method. The Investigators were not blinded to allocation during experiments and outcome assessment unless explicitly stated for the method.

### Plasmids
The ectodomain and full-length version of Ten3-$A_0B_0$ (MGC premier cDNA clone BC145284, Biocat GmbH, BamH1 sites removed with silent mutations) were cloned into the pUPE106.03 and pUPE3820 vectors (U-protein Express), containing a N-terminal cystatin secretion signal followed by a hexa-His tag or a N-terminal GFP tag, respectively, by using BamHI and NotI restriction sites. Ectodomains and full-length splice variants of Ten3-$A_0B_1$, Ten3-$A_1B_0$, Ten3-$A_1B_1$ were constructed from full-length Ten3-$A_0B_0$ using PmlI-AgeI, EcoRI-PmlI and EcoRI-AgeI restriction ligation cloning. Monomeric mouse Ten3 (residues 846-2715) for stripe assays as in Jackson et al.[23].

## Protein expression and purification

Epstein−Barr virus nuclear antigen I-expressing HEK293 cells (HEK-E; U-Protein Express) were used to express all secreted Ten3 ECD isoforms. Cells were cultured in FreeStyle293 expression medium with GlutaMAX (FreeStyle; Gibco) supplemented with 0.2% foetal bovine serum (FBS, Gibco) and 0.1% Geneticin (G418 Sulfate; Gibco) in a shaking incubator at 37 °C and 5%. Prior to transfection, HEK-E cells were seeded at $0.25\text{-}0.32 \times 10^{6}$ cells/mL onto 1L Erlenmeyer cell culture flasks in FreeStyle medium without supplements. 24h later, cells were transfected with a total of 125 µg DNA encoding the full ECD per 250 mL of cells using polyethylenimine (PEI, 1:3 DNA: PEI ratio; Polysciences), according to the manufacturer's protocol, and treated with 5.5% Primatone in Freestyle medium 6−24h post-transfection. Six days after transfection, the medium was collected by pelleting the cells through centrifugation at 300 g for 10 min. The medium was once again centrifuged at 4000 g for 10 min to remove any remaining cells from the medium. Proteins were purified by Ni-NTA affinity chromatography using an elution buffer containing 25 mM HEPES (pH 7.8), 500 mM NaCl, and 500 mM Imidazole, followed by size-exclusion chromatography (SEC) using a Superose6 Increase 10/300 GL column (Cytiva) into a final buffer composition (SEC-buffer) of 20 mM HEPES (pH 7.8), 150 mM NaCl. After each Ni-NTA affinity and SEC purification step, the proteins were concentrated using a 15 mL Amicon ultra centrifugal filter at 100 kDa cut-off. The final SEC-buffer was either supplemented with 2 mM $CaCl_2$ or concentrated proteins were dialysed overnight at 4 °C using 3.5K MWCO Slide-A-Lyzer MINI Dialysis Devices (Thermo Scientific) in 100 mL SEC buffer containing 2 mM $CaCl_2$. Purified proteins were diluted to 20 µg/mL, boiled for 5 min at 98 °C in the presence or absence of β-mercaptoethanol and loaded onto a TGX mini protean precast gel (Biorad) for 50 min at 200V.

## SEC-MALS

The oligomeric and conformational states of the different purified isoforms were probed and directly compared by using analytical size exclusion chromatography with subsequent multi-angle light scattering (SEC-MALS). The Ten3 isoforms, along with bovine serum albumin (BSA) reference, were diluted to a final concentration of 0.5 mg/mL in SEC buffer (20 mM HEPES pH 7.8, 150 mM NaCl.) supplemented with 2 mM $CaCl_2$ and loaded onto a Superose6 Increase 10/300 GL column (Cytiva) integrated on a high-performance liquid chromatography (HPLC) unit (1260 Infinity II, Agilent) with an online UV detector (1260 Infinity II VWD, Agilent), an 8-angle static light scattering detector (DAWN HELEOS 8+, Wyatt Technology), and a refractometer (Optilab T-rEX, Wyatt Technology) in series. On the basis of the measured Rayleigh scattering at different angles and the established differential refractive index increment of value of 0.185 ml g$^{-1}$ for proteins in solution with respect to the change in protein concentration (dn/dc), weight-averaged molar masses for each species were calculated using ASTRA software (Wyatt Technology; v.7.3.1).

## Cryo-electron microscopy and data processing

All cryo-EM imaging of Ten3 ECDs was performed on purified protein samples diluted in SEC-buffer supplemented with 2 mM $CaCl_2$ to a final concentration of 0.5 mg/mL. 3.0 µL of diluted sample was deposited onto Quantifoil R1.2/1.3 Cu 300 mesh grids that were glow-discharged for 1 minute at 15 mA. Excess sample volume was blotted away for 2.5−4.0 seconds with blot force of -3.0 at 22 °C and 100% humidity before plunging into liquid ethane using a Vitrobot Mark IV (Thermo Scientific). Movies were acquired using a K3 detector (Gatan) in counting super-resolution mode with a 20 keV (Gatan) slit width at 105,000x nominal magnification, a corresponding super-resolution pixel size of 0.418 Å, and a total dose of 50 e⁻/Å² per movie. A total of 10,927; 3,888; 4,412 and 8,447 movies were acquired for the $A_0B_0$, $A_0B_1$, $A_1B_0$, and $A_1B_1$ isoforms, respectively, at a defocus range of -0.8 to -2.0 µm (-0.8 to -2.3 µm for the $A_0B_0$ isoform). All datasets were processed using RELION3.1[54] unless otherwise stated. Beam-induced motion and drift correction was performed using MotionCorr2[55] with a pixel binning factor of 2 (0.836 Å physical pixel size), and Ctf estimation by using Gctf[56]. Particles were initially picked from the corrected images using RELION's own Laplacian-of-Gaussian method, upon which a manual selection from subsequent 2D classification was used for reference-based particle re-picking. 2D classification was used to clean data of non-particle objects like foil hole edges and ice crystals from the extracted particle sets. Random subsets of particles were further 2D classified to select class averages for ab initio reconstruction of a reference for 3D classification on the full set of particles that were cleaned through 2D classification. Automated particle picking of $A_1B_0$ compact dimers was performed in RELION using the $A_1B_1$ compact dimer refined map as a reference after 20 Å low-pass filtering. For all datasets, 3D map reconstructions were performed according to the regular workflows of Relion3.1, and as reported in the corresponding Supplementary Figs. C2-to-C1 symmetry expansions for the $A_1B_1$ compact dimer were performed on refined particle datasets. Particle subtraction was performed without automated centreing of the resulting maps. For particle subtraction in symmetry-expanded data, the C2-to-C1 particles were first refined with a full-map reference in C1 before masking regions of interest on a single subunit for subtraction outside the masked region. To select particles within datasets that contained good-quality signal for the EGF domains and splice insert A, focussed 3D classification was performed with increased regularization parameters T, while masking the region of interest. To visualize the compact dimer movement from open to closed conformation, 3D variability analysis in CryoSPARC[57] was used. CryoSPARC's local refinement was used for the reconstruction of the dimer particles containing splice insert A density.

## Model building and refinement

Monomeric Ten3-$A_0B_0$ (6FAY) was combined with models predicted by ColabFold[58] for EGF6-8 ($A_0$), C-rich, ABD, and Tox-GHH domains. Non-compact subunit models for the other isoforms were created by predicting structures for the EGF6-8 containing splice insert A and NHL containing B and substituting them in the initial $A_0B_0$ model. The Ten3 'stalk' was predicted by ColabFold multimer for a dimeric amino acid input spanning the residues of the ECD through EGF5, which exclusively resulted in stalks with EGF2-2/EGF5-5 configuration. To model the compact dimer, two $A_1B_1$ subunit models were refined in the compact dimer closed conformation full map by iterative cycles of real space refinement in phenix.real_space_refine of the Phenix package[59] and manual real-space refinement in Coot[60]. EGF6-7 and A, which were outside the map, were removed from the model. Residues and side chains lacking sufficient density were removed from the model to enhance the refinement process. At the final stages, the model was refined in the full map of teneurin-3 $A_1B_1$ compact dimer using C2 symmetry constraints in Phenix (Supplementary Data 1), and remaining Ramachandran outliers were solved manually in Coot. A single subunit $A_0B_0$ model containing EGF6 through the C-terminal Tox-GHH domain was refined in the map in which EGF6 and 7 were additionally resolved. Before refinement, this model's C-rich domain was substituted for the C-rich domain from the final refined $A_1B_1$ compact dimer model. The stereochemistry of the models was checked with MolProbity[61], the interfaces were analysed with the Protein Interfaces, Surfaces and Assemblies (PISA) web server[62]. The structural figures of maps and models were generated with Chimera[63] and PyMOL (Schrodinger Molecular Graphics System, DeLano Scientific, San Carlos, CA). Protein sequence alignments were generated using Clustal Omega[64].

## Small-angle X-ray scattering

SAXS was performed at the European Synchrotron Radiation Facility BM29 beamline, at 12.5 keV operating energy equipped with a Pilatus3

2M detector. Samples at approximately 1 mg/mL were diluted two-fold twice in series, and then 50 microliters of each were injected, i.e., at ~1, ~0.5 and ~0.25 mg/mL for each of the four Ten3 splice variants and at three buffer conditions. The three conditions, all containing SEC buffer (150 mM NaCl, 20 mM HEPES, pH 7.8), were either without additional calcium, with 2 mM $CaCl_2$ or with 5 mM EDTA added. Data were collected at 20 °C over 10 frames, one frame per second, with a scattering vector range of 0.0456–5.15 nm$^{-1}$. Images were radially averaged and normalised using the BM29 EDNA-based pipeline[65]. Radiation damage was monitored, and data frames were selected manually in the PRIMUS GUI[66], which was also used for frame averaging, buffer subtraction, Guinier modelling, and determining the pair-distance distribution P(r). Kratky analyses were performed according to Durand et al. [67]. In preparation for rigid-body fitting, complex glycan trees were built manually on glycosylation sites of the compact models in Coot[60]. A model for the N-terminal domains that could not be resolved by cryo-electron microscopy was generated in ColabFold[58]. These were then taken together, and short flexible linkers were introduced between domains, conserving the interfaces of the compact dimers, as well as the EGF2/EGF5 covalent dimerisation interface, and fit to the datasets of each splice insert was optimised using CORAL[68]. The fit of each resulting model to each dataset was then calculated in CRYSOL[69], using 50 as the "number of spherical harmonics" and allowing for constant subtraction. All programs were used as implemented in ATSAS 3.1.3[70]. SAXS measurements of protein dilutions for which insufficient coherent frames were collected to reliably calculate an averaged scattering curve were excluded, with a maximum of one exclusion per protein sample.

## Thermal stability assay
Thermal shift assay (TSA) assays were performed using purified human Ten3 isoforms at a concentration of 0.5 mg/mL. Prior to the experiments, the samples were buffer exchanged into SEC buffer (150 mM NaCl, 20 mM HEPES, pH 7.8) without additional calcium denoted "SEC" condition, into SEC buffer supplemented with 2 mM $CaCl_2$ (physiological calcium) denoted "$Ca^{2+}$" condition, or into SEC buffer supplemented with 5 mM EDTA (no calcium) denoted "EDTA" condition. SYPRO Orange Dye (Invitrogen) was diluted to a concentration of 5X concentrated solution and filtered using a 0.22 μm membrane. Final protein concentrations were 60 μg/mL protein, 1× dye and final buffer concentrations 132 mM NaCl, 17.6 mM HEPES, pH 7.8, supplemented with 1.76 mM $CaCl_2$ or EDTA. A temperature ramp from 5 °C to 95 °C was set up at a speed of 0.02 °C/s on a QuantStudio 3 Real-Time PCR system (Thermo Fisher Scientific). All measurements were performed in triplicates. The melting temperatures $T_m$ were determined as the intersection of the x-axis with the smoothened second derivative of the melting curve between the valley and peak shown in the raw melting curves.

## Negative stain imaging and analysis
Purified protein samples were diluted to 20 μg/mL before depositing 3.5 μL of the resulting sample volume onto glow-discharged carbon-coated copper grids for 2 min incubation. Excess sample volume was then blotted away, followed by three washing cycles of depositing 10 μL of SEC buffer supplemented with 2 mM $CaCl_2$ with subsequent blotting. The grid was then incubated with 3.5 μL 2% uranyl acetate or 1% uranyl formate (for $A_0B_0$) for another 60 sec before blotting the excess away. Grids were left to dry at RT for up to 2 min before imaging or storage. At least 20 micrographs were acquired for each sample using a JEOL JEM-1400Plus transmission electron microscope operating at 120 kV with a standard magnification of 30,000x, 1000 ms exposure, and a defocus value of -2.4 μm (or, Tecnai12 microscope operated at 120 kV for $A_0B_0$). Micrographs were processed using RELION3.1[54] before following standard reconstruction workflow. In short, particles were first picked using RELION's own Laplacian-of-Gaussian implementation, followed by reference-based particle picking after 2D classification. An initial 3D model was generated for multiple rounds of 3D classification of all particles. The final selection was refined and postprocessed in RELION. To determine model-map correlations, two non-compact subunits of the isoform corresponding to the nsTEM electron densities (with EGF6-8 for $A_0$ and without for $A_1$ isoforms) were fit using "Fit in map" tool in Chimera. A calculated low-resolution filtered map (molmap) of 20 Å resolution was created from each dimeric model and fit to all experimental densities using "Fit in map" to acquire correlation values.

## Cell clustering assay
Cell electroporation for the clustering assays was performed as previously reported[71]. K562 cells (catalogue nr. ACC 10, Leibniz Institute DSMZ) were cultured in RPMI-1640 medium (Gibco), supplemented with 10% FBS (Gibco) and 1% Penicillin/Streptomycin (Gibco), and grown in a shaking incubator at 37 °C and 5% $CO_2$. Prior to electroporation, K562 cells were centrifuged for 5 min at 300 g and washed in 1x PBS (Gibco). Cells were once again centrifuged for 5 min at 300 g and resuspended in buffer R (Gibco). Per condition, $2 \times 10^6$ cells were incubated with a total amount of 15 μg of DNA (Ten3:empty vector ratio was 1:5) for 15 min at room temperature. After the incubation, K562 cells were electroporated with the Neon Transfection System (Thermo Fisher Scientific), using the following parameters: 1450 V, 10 ms pulse length, and 3 pulses[7]. After electroporation, cells were directly seeded randomly onto 5 mL of pre-warmed RPMI-1640 medium with 10% FBS in 6-well plates. Cells were placed in a shaking incubator at 37 °C and 5% $CO_2$ for approximately 20 h. Cells were imaged blindly on an EVOS M5000 microscope with a 10x objective (0.25 NA; EVOS, Thermo Fisher Scientific) using the EVOS LED GFP cube (Thermo Fisher Scientific). The researcher was blind to which condition was being imaged to prevent bias for cluster formation. In the GFP channel, regions of interest (ROIs) larger than 100 pixels were identified using Fiji[72] Analyse Particles after rolling ball background subtraction with a 50-pixel radius was performed. The area of each ROI was then measured. A cell cluster was defined as an object three times larger than the mean large single-cell size (800 pixels). The clustering index was determined as the summed cluster area divided by the summed area of all ROIs (clusters + non-clusters) times 100%. The cluster size and clustering index were averaged per image, and data from three independent experiments (5 images per experiment, 15 images total per condition). Statistical significance was determined by performing a one-way ANOVA followed by Tukey's multiple comparison test. All analysed data are represented as mean ± SEM.

## Animals for neuronal assays
Fertilised chicken eggs (*Gallus gallus*, white leghorn) were obtained from a commercial dealer and incubated in a forced draught incubator at 37 °C and 65% humidity until the desired embryonic stage.

## In Situ Hybridisation (ISH)
Eyes from chick embryos (*Gallus gallus*, White Leghorn strain, obtained from Henry Steward Ltd.) at developmental day (E) 6 were fixed in 4% paraformaldehyde (PFA) solution for 2 days, followed by perfusion with 4% sucrose/0.5% PFA solution, until samples sunk to the bottom of the tube. Eyes were embedded in O.C.T. cryo-embedding matrix prior to cryo-sectioning. 20-μm thick coronal sections were cut on the cryostat to obtain dorsoventral sections of the chick retina and stored at -80 °C. Sections were rehydrated for 5 min in PBS-T (1x PBS + 0.1% Tween-20), and permeabilised by digestion with 5 μg/mL Proteinase K for 5 min at room temperature. Digestion was inhibited by 2mg/mL (2%) glycine in PBS-T, followed by 2 times 5 min washes with PBS-T and fixation with 4% PFA for 20 min. Residual PFA was removed by washing twice with PBS-T for 5 min, and sections were pre-hybridised in hybridisation mix (50% formamide, 5x SSC pH 4.5,

50 µg/mL tRNA, 1% SDS and 50 µg/mL Heparin) for 1 hour at 65 °C, prior to hybridisation of probe overnight at 65 °C. Sections were washed 3 times for 15 min with a solution containing 50% formamide, 5 X SSC pH 4.5 and 1% SDS and 3 times for 15 min in solution comprising 50% formamide, 2 X SSC pH 4.5, at 65 °C and 60 °C, respectively.

After blocking for 1 hour at room temperature with 10% sheep serum in TBS-T (1x TBS + 0.1% Tween-20), sections were incubated with polyclonal Alkaline Phosphatase (AP)-conjugated anti-DIG antibody (Roche, 11093274910) diluted 1:2000 in 10% sheep serum in TBS-T overnight at 4 °C. Samples were extensively washed to remove excess antibody. Sections were then washed thrice for 10 min in NTMT buffer containing 100 mM NaCl, 100 mM Tris-HCl pH 9.5, 100 mM $MgCl_2$ and 1% Tween-20. The colour staining was developed at room temperature and in the dark using NBT/BCIP stock solution (75 mg/mL and 50 mg/mL, respectively) (Roche) diluted 1:200 in NTMT buffer. The reaction was terminated by 2 washes for 10 min in NTMT, followed by a wash in PBS-T pH 5.5 for 10 min and 2 washed for 10 min in 1x PBS. Samples were postfixed in 4% PFA in PBS for 30 min at room temperature and stained with Hoechst diluted 1:10,000 in 1x TBS before a final wash in PBS and mounting in 90% glycerol.

### Stripe assays

Stripe assays were performed according to a previously established protocol[41]. Prior to the experiment, specially designed silicone matrices were boiled in autoclaved water, glass coverslips were coated in 10 µg/mL poly-L-lysine (PLL) for 1 hour, and both were left to dry overnight in a laminar flow hood. On the day of the experiment, dry matrices were fixed onto glass coverslips. Solutions were applied using a Hamilton syringe which was rinsed before use with ethanol and Hank's Basic Salt Solution (HBSS). Monomeric $Ten3-B_0/B_1$ proteins were diluted in HBSS (ThermoFisher) and used at a final concentration of 10 µg/mL. Before application, monomeric $Ten3-B_0/B_1$ was clustered using monoclonal anti-6xHis Tag antibodies (GeneTex, GTX44514) and fluorescently labelled by adding a small amount of polyclonal goat anti-rabbit AlexaFluor 568 secondary antibody (ThermoFisher, A-11011) enabling subsequent identification of 1[st] and 2[nd] applied stripes. The anti-6xHis Tag antibody (10 µg/mL) alone was used for generating control stripes (2[nd] stripe). Laminin and merosin solutions were prepared to final concentrations of 5 µg/µL and 1 µg/mL, respectively. Neurobasal (NB) medium was supplemented with 2% methylcellulose, 1% B27, 1% glutamine, 1% Penicillin/Streptomycin (Pen/Strep) and 0.01% forskolin.

To generate the first set of stripes, clustered purified protein solutions were injected into the matrices using a Hamilton syringe and culture dishes were incubated at 37 °C/5% $CO_2$ for 45 min. Following the removal of matrices from the coverslips and HBSS wash, an unlabelled second stripe solution containing the anti-6xHis Tag antibody in HBSS was pipetted on top of the stripes. Culture dishes were incubated for 45 min at 37 °C, and coverslips were subsequently coated with laminin/merosin solution for 1 hour. Freshly dissected chick retinas at embryonic stage E6 were flat-mounted on nitrocellulose filter paper and cut into ~2 mm wide strips along the dorsoventral axis using a tissue chopper. Explant strips were placed perpendicularly on top of protein stripes and incubated at 37 °C/ 5% $CO_2$ for up to three days in neurobasal media. Explant strips were fixed in 4% PFA/0.33% sucrose for 15–20 min at room temperature and permeabilised by washing twice for 5 min with 1x PBS/0.1%Triton and subsequently stained by incubation with 1:200 phalloidin conjugated with AlexaFluor 488 for 20 min at room temperature, in a wet chamber and under aluminium foil. After a series of washes to remove the excess staining solution, coverslips were mounted onto glass slides using Mowiol. Stripe assays were imaged using a Fluorescence microscope (Zeiss) at 20x magnification. Three independent rounds of stripe assays were performed, each round consisting of 4–6 individual assays per condition (Control vs monomeric $Ten3-B_0/B_1$), using explants from at least 2 animals. For individual stripe assay, two images were taken from the ventral and dorsal regions of each retinal explant strip. Axon guidance decisions in all images were scored blindly by 3 people, in accordance with a previously established scoring system, with values between -3, 0 and +3, representing strong repulsion, no guidance and strong attraction, respectively[41,42]. Mean scores were calculated and plotted using GraphPad (mean ± SEM), and statistical significance was determined with Two-way ANOVA with Tukey's multiple comparisons tests.

### Inclusion and diversity statement

In line with our commitment to promoting a diverse and inclusive scientific community, we affirm our dedication to upholding the principles of equity and diversity in the research presented in this paper. We believe that diversity in the scientific community is essential to drive innovation, foster creativity, and ensure comprehensive and unbiased exploration of the life sciences.

### Reporting summary

Further information on research design is available in the Nature Portfolio Reporting Summary linked to this article.

## Data availability

The structure model data generated in this study have been deposited in the Protein Data Bank (PDB) under the accession codes 8R50 (teneurin-3 $A_1B_1$ compact dimer), 8R51 (teneurin-3 $A_1B_1$ non-compact subunit), and 8R54 (teneurin-3 $A_0B_0$ non-compact subunit). Previously published structure model data used in this study can be found in the PDB under the accession codes 6FAY (teneurin-3 $A_0B_0$ non-compact subunit), 7BAM (teneurin-4 compact dimer), 6SKA (teneurin-2 in complex with latrophilin-1), and 5FTU (latrophilin-3 in complex with FLRT2). The cryo-EM density map data generated in this study have been deposited in the Electron Microscopy Data Bank (EMDB) under the accession codes EMD-18889 (teneurin-3 $A_1B_1$ compact dimer), EMD-18890 (teneurin-3 $A_1B_1$ non-compact subunit), EMD-18891 (teneurin-3 $A_0B_0$ non-compact subunit), EMD-18900 (teneurin-3 $A_0B_1$ non-compact subunit), EMD-18902 (teneurin-3 $A_1B_0$ non-compact subunit), EMD-19409 (teneurin-3 $A_1B_0$ compact dimer). See Supplementary Data 1 for specifications. The SAXS data generated in this study have been deposited in the Small Angle Scattering Biological Data Bank (SASBDB) under the accession codes SASDTY2 (0.97 mg/mL teneurin-3 $A_1B_1$), SASDTZ2 (1.24 mg/mL teneurin-3 $A_1B_1$ with calcium), SASDT23 (0.62 mg/mL teneurin-3 $A_1B_1$ with calcium), SASDT33 (0.31 mg/mL teneurin-3 $A_1B_1$ with calcium), SASDT43 (1.30 mg/mL teneurin-3 $A_1B_1$ with EDTA), SASDT53 (0.44 mg/mL teneurin-3 $A_0B_0$), SASDT63 (0.46 mg/mL teneurin-3 $A_0B_0$ with calcium), SASDT73 (0.23 mg/mL teneurin-3 $A_0B_0$ with calcium), SASDT83 (0.65 mg/mL teneurin-3 $A_0B_0$ with EDTA), SASDT93 (0.65 mg/mL teneurin-3 $A_0B_1$), SASDTA3 (1.43 mg/mL teneurin-3 $A_0B_1$ with calcium), SASDTB3 (0.72 mg/mL teneurin-3 $A_0B_1$ with calcium), SASDTC3 (0.36 mg/mL teneurin-3 $A_0B_1$ with calcium), SASDTD3 (0.62 mg/mL teneurin-3 $A_0B_1$ with EDTA), SASDTE3 (0.69 mg/mL teneurin-3 $A_1B_0$), SASDTF3 (1.44 mg/mL teneurin-3 $A_1B_0$ with calcium), SASDTG3 (0.74 mg/mL teneurin-3 $A_1B_0$ with calcium), SASDTH3 (0.36 mg/mL teneurin-3 $A_1B_0$ with calcium), SASDTJ3 (0.84 mg/mL teneurin-3 $A_1B_0$ with EDTA). See Supplementary Data 2 for specifications. Source data are provided with this paper.

## Code availability

Scripts and code used for the K562 clustering analysis used in this study are available through Zenodo [https://doi.org/10.5281/zenodo.10843181].

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

## Acknowledgements

Cryo-EM data were collected at The Netherlands Centre for Electron Nanoscopy (NeCEN) with assistance from Willem Noteborn. We acknowledge the European Synchrotron Radiation Facility (ESRF) for provision of synchrotron radiation facilities, and we would like to thank Petra Pernot for assistance and support in using beamline BM29. UD is supported by a BBSRC grant (BB/T013753/1). BJCJ is supported by an NWO grant (OCENW.KLEIN.026). DHM is supported by an NWO computing grant (2021.058) and NWO Veni grant (722.016.004).

## Author contributions

Conceptualisation: C.G. and D.H.M. Methodology: C.G., J.W.B., C.P.F., N.J., and U.D. Data acquisition and analysis: C.G., J.W.B., C.P.F., N.J., and L.K. Supervision: R.H., B.J.C.J., and D.H.M. Writing – original draught: C.G., J.W.B., C.P.F, B.J.C.J., R.H., and D.H.M. Writing – review & editing: C.G., J.W.B., C.P.F., N.J., L.K., B.J.C.J., R.H., and D.H.M.

## Competing interests

The authors declare no competing interest.
