## [Peer Review File · Nature Communications]

REVIEWER COMMENTS

Reviewer #1 (Remarks to the Author):

This is a great piece of structural biology work which includes functional aspects, centered on how two splice inserts control and modulate the dimer formation and function of Ten3. The role of alternative splicing in teneurins is key to their trans-cellular interactions, yet the structural basis of splicing-dependent assembly formation remains unclear. The authors describe important mechanistic details that help explain the function of this neuronal protein in synapse formation and the possible biological links to specific human diseases. The group employs state of the art technologies and the writing of the manuscript is clear. Several orthogonal techniques (Cryo EM, SAXS, Single particle negative stain, trans-cellular complex formation, etc.) are used to obtain robust data that can be interpreted properly.

Overall I have one major issue regarding the presentation of the SAXS data. As the authors were unable to use CryoEM (air-water interface issue) to resolve the constructs with and without the A and B splice sites, they turned (rightly in my opinion) to SAXS. The data that the authors chose to present however are only Kratky plots and 3D envelopes. While these are interesting piece of data, much more information (and necessary standard controls) can be squeezed out of the same datasets.

I suggest two things: first, that the authors present both scattering curves (raw data) and the linear Guinier profiles. These are required datasets that describe the quality of the data. These bits of data could go into supplemental figures but they will give the confidence to the reader (and this reviewer) to know that the SAXS data are suitable to generate 3D models, just like the authors did in their Ten 4 paper in 2022.

The second advice is that reliable information can be obtained by generating (and showing) the pair-wise distribution (Pr) profile and the accompanying Dmax. Kratky are a bit limited to show folding compactness (e.g. an extended Kratky indicates unfolding, but not necessarily a dynamic change in subdomain positions. The Pr are more apt at showing changes in size and shape of the protein with or without the splice inserts.

While they are at it, I suggest they follow the guidelines reported below when reporting SAXS datasets. I see that the datasets have been deposited but I do not have access at this moment and it would be easier for the reader to have this information somewhere in the paper.

Table 5 of this <https://journals.iucr.org/d/issues/2017/09/00/jc5010/jc5010.pdf>

Or <https://journals.iucr.org/d/issues/2017/09/00/jc5010/index.html#BB32>

Minor issues:

ABSTRACT. The last sentence states that the alternative splicing of the Ten3 “influencing [...] circuit wiring”. I would add “likely influences circuit wiring”. It may happen, but to be fair none of the experiments presented here directly test circuit formation in vivo. Alternative splicing could influence, say, membrane recycling or other subtle features of the synapse but not necessarily circuit formation.

INTRODUCTION, PAGE 2: “Finally, all members of the teneurin family have been associated [...] disorders” should be written as “Finally, (GENETIC ABNORMALITIES OF) all members of the teneurin family have been associated [...] disorders” or something to this effect. It’s not the native protein that cause autism or essential tremors but some abnormality of it.

PAGE 4, RESULTS: To examine the structural basis [...] the complete extracellular domain (ECD, Δ1-341 residues). I suggest to define this construct as ECD, 342-2715. First, it will indicate what it is, rather than what it isn’t, and second, it will give an immediate idea of the size, as I was initially confused by thinking that Ten3 was only 341 residues long.

PAGE 4, RESULTS: The interface is mainly composed of hydrophobic and -phobic surface. What does this (-phobic) mean?

PAGE 4, RESULTS: numbering is probably wrong
the EGF8 domain (residues 762-764). Only 3 residues?
ABD domain (residues 2595-2597) Only 3 residues?

PAGE 5: The main difference [...] of the two SUBUNITS with respect to each other (Fig. 3A)”. I suggest to use the term monomer or protomer instead of SUBUNIT, as in such a large and multidomain protein subunit may lead to some confusion.

BOTTOM OF PAGE 5: In absence of splice insert A (A0B0 and A1B1 isoforms). Should that be A0B1?

In addition to ref 42, this one should be cited as well

Lloyd BA, Han Y, Roth R, Zhang B, Aoto J. Neurexin-3 subsynaptic densities are spatially distinct from Neurexin-1 and essential for excitatory synapse nanoscale organization in the hippocampus. *Nat Commun.* 2023 Aug 5;14(1):4706. doi: 10.1038/s41467-023-40419-2. PMID: 37543682; PMCID: PMC10404257.

Reviewer #2 (Remarks to the Author):

Synapse formation and elimination are intricately orchestrated by an array of cell-adhesion molecules (CAMs), essential in cell-cell interactions and the construction of neuronal networks. Among these CAMs, teneurins are recognized as key recognition molecules, crucial for enabling trans-synaptic communication. Teneurins are characterized by alternative-splicing isoforms, whose unique shapes play a pivotal role in molecular recognition. This influences the binding specificity of partners and plays a key role in synaptic differentiation. Specifically, mouse teneurins-3 has two splice insertion sites: one between EGF7 and EGF8 (splice insert A), and another between the first and second blade of the NHL β -propeller (splice insert B). In this study, Gogou et al. utilized a trio of techniques: Cryo-Electron Microscopy (Cryo-EM), X-ray scattering (SAXS), and negative stain EM, to decipher four structures of mouse teneurin-3 homodimer complexes, each with distinct splice variants combinations. The team first resolved the structure of a compact teneurin-3 A1B1 dimer using Cryo-EM and further investigated the unpacked versions of all four splice isoforms, integrating SAXS and negative stain EM to summarize the dimer models of each splice variant. Subsequent clustering assays assisted them to model higher-order interactions formed by dimers of dimers. Finally, stripe assays revealed that splice insert B alone can guide axon growth. This research offers crucial insights into how short splice inserts modify the conformation of teneurins, altering their dimerization formats and exposing distinct protein surfaces for binding with other proteins. This forms the foundation for specific cellular interactions. Overall, the structures presented in this study are of high quality and provide important insights into how splicing-dependent assembly formations contribute to cell-specific recognition and the development of the neuronal network. While the work is clearly presented and well-executed, there are some major concerns.

Major concerns:

1. While the A1B1 structure of teneurin-3 is derived directly from high-resolution Cryo-EM, the other three dimer models (A1B0, A0B0 and A0B1) are constructed from indirect data obtained through SAXS and negative stain EM. Analysis of Figure 5 suggests that most of the model aligns well with the low-resolution 3D reconstructions, yet some uncertainty remains due to the low resolution of the map. It would be beneficial if the authors could provide additional analysis to validate the accuracy of the proposed dimer assembly formats, thereby minimizing ambiguity stemming from the low resolution.
2. The study identifies four divergent structural assemblies arising from different splice variants of mouse teneurin-3. However, it remains to be demonstrated whether these conformations exist within an animal and their physiological relevance. The authors are encouraged to provide evidence, such as some mutagenesis or methodologies that could affirm the physiological significance of these conformations.

3. The paper briefly outlines the binding modes of isoform-specific teneurin-latrophilin trans-cellular complexes. Given that FLRT is also involved in transsynaptic interactions, it would be valuable if the authors could expand on their predictions or descriptions regarding how FLRT interacts with the teneurin-latrophilin complex. Such information would deepen the understanding of these complex molecular interactions.

Minor points:

1. Regarding Figure 4B, I suggest adjusting the color representation of EGF8 in the density maps of A0B0 and A0B1 to blue. This change would ensure consistency in the depiction of EGF8 across Figures 4B, 4D, and 4E, making it easier for readers to follow and compare the features across these figures.
2. In Figure 6C and Supplementary Figure 5B, the current map format lacks clarity, making it challenging to discern the conformation and interaction details. Specifically, it's difficult to identify the localization of each subunit and the contact points between the two dimers. To improve this, I recommend that the author consider revising them to incorporate either the model alone or a combination of the model with a transparent overlay of the map. It would likely provide a clearer and more detailed visualization of the assembly, facilitating a better understanding of the structures and interactions depicted.

Reviewer #3 (Remarks to the Author):

The manuscript entitled "Alternative splicing controls teneurin-3 compact dimer formation for neuronal recognition" by Gogou et al, addresses an important aspect of Neuronal network formation by looking at alternative splicing structural consequences on the teneurin family of cell adhesion molecules, by determining cryoEM structures and biophysical and functional assays with Teneurin3. That way, it offers insights into homo- and heterophilic interactions and thus on neuronal recognition and circuit wiring.

The paper is extremely well-written and is easy to follow. Moreover, the figures are carefully displayed and clearly illustrate the results and conclusions. There are however some aspects that could still be improved, such as:

Lines 48 to 50. If all members of the teneurin family have been associated with neurological disorders, why not name one to teneurin-2? Since all other 3 are illustrated with examples.

-

1) Lines 51 to 58. The author did a good job describing the domain organization of Ten3 but left the intracellular domain (ICD) out. It should be introduced, and if relevant literature exists, it should be cited.

2) Calcium is used for protein purification and tested in the melting temperature assays and SAXS, but its influence on the structure/function of the protein is not adequately described in the introduction. In my opinion the authors should add a bit of information on this matter.

3) Line 156. It should be A0B0 and A0B1

4) Line 175 should be "...and supplemental Fig. 5B"

5) Figure 6C- In the panel on the right, the 2 transmembrane helices overlap. Is this a mistake or it reflect the position after rotation of the full dimer, and the TM are one in front of the other, from the viewpoint of the reader?

6) Figure 7 A – are both scale bars 400 um? Should not the II-Hoechst represent a smaller value?

7) In methods and on the "Stripe assays" is written "Ten3-B0/B1 monomeric (referred to as Ten3 from now on)". If there were many entries after it was understandable, but since it is named only twice after, I think using the full description "Ten3-B0/B1 monomeric" would be better.

Minor things:

Line 389. A space is missing

Line 426 A1B1, the numbers should be below the line (subscript). It happens twice in the same line.

Line 436 A1B1 (see above)

Line 483 a space too much (20- mM)

Line 517 & 575 CO₂ – the 2 should be subscript

Line 549 MgCl₂ – the 2 should be subscript

REVIEWER

COMMENTS

Reviewer #1 (Remarks to the Author):

This is a great piece of structural biology work which includes functional aspects, centered on how two splice inserts control and modulate the dimer formation and function of Ten3. The role of alternative splicing in teneurins is key to their trans-cellular interactions, yet the structural basis of splicing-dependent assembly formation remains unclear. The authors describe important mechanistic details that help explain the function of this neuronal protein in synapse formation and the possible biological links to specific human diseases. The group employs state of the art technologies and the writing of the manuscript is clear. Several orthogonal techniques (Cryo EM, SAXS, Single particle negative stain, trans-cellular complex formation, etc.) are used to obtain robust data that can be interpreted properly.

Overall I have one major issue regarding the presentation of the SAXS data. As the authors were unable to use CryoEM (air-water interface issue) to resolve the constructs with and without the A and B splice sites, they turned (rightly in my opinion) to SAXS. The data that the authors chose to present however are only Kratky plots and 3D envelopes. While these are interesting piece of data, much more information (and necessary standard controls) can be squeezed out of the same datasets.

I suggest two things: first, that the authors present both scattering curves (raw data) and the linear Guinier profiles. These are required datasets that describe the quality of the data. These bits of data could go into supplemental figures but they will give the confidence to the reader (and this reviewer) to know that the SAXS data are suitable to generate 3D models, just like the authors did in their Ten 4 paper in 2022. The second advice is that reliable information can be obtained by generating (and showing) the pair-wise distribution (Pr) profile and the accompanying Dmax. Kratky are a bit limited to show folding compactness (e.g. an extended Kratky indicates unfolding, but not necessarily a dynamic change in subdomain positions. The Pr are more apt at showing changes in size and shape of the protein with or without the splice inserts.

While they are at it, I suggest they follow the guidelines reported below when reporting SAXS datasets. I see that the datasets have been deposited but I do not have access at this moment and it would be easier for the reader to have this information somewhere in the paper.

Table 5 of this <https://journals.iucr.org/d/issues/2017/09/00/jc5010/jc5010.pdf>

Or <https://journals.iucr.org/d/issues/2017/09/00/jc5010/index.html#BB32>

We would like to thank the reviewer for the compliments on the structural work and the suggestions to improve our reporting of the SAXS data. We have followed up on the reviewer's suggestions in the revised manuscript and added the scattering curves, linear Guinier plots, Dmax values and a new Table 2 reporting the SAXS data in detail. More specifically, raw data and Guinier plots from the SAXS data are added as Supplementary Figure 5A and D respectively. Pair-distance distribution profiles for all SAXS data are added to Supplementary Figure 5C and for the calcium-containing samples additionally as a new panel B to Figure 5. Dmax values for the four splice variants in calcium containing conditions are reported in Figure 5B and for all SAXS data in Table 2. The data-to-model fit in the Guinier region supports the quality of the SAXS

data (Supplementary figure 5D and Table 2) and the pair-distance distributions and Dmax values support our analysis of the SAXS data. The manuscript text has been updated accordingly to refer to the added data.

Additionally, we would like to thank the reviewer for sharing the guidelines for reporting SAXS datasets and we have updated the information on sample details, data acquisition and data presentation/methods/analysis in the method and results section of the paper. Specifically, we have added Table 2 as an overview, along with specific per-graph descriptions in Supplemental Fig. 5.

Finally, please see here the list of passwords to access the SAXS data. The data will be made public upon publication of the article: <https://www.sasbdb.org/project/2165/h02uirl8r7/>

Minor issues:

ABSTRACT. The last sentence states that the alternative splicing of the Ten3 “influencing [...] circuit wiring”. I would add “likely influences circuit wiring”. It may happen, but to be fair none of the experiments presented here directly test circuit formation in vivo. Alternative splicing could influence, say, membrane recycling or other subtle features of the synapse but not necessarily circuit formation.

We thank the reviewer for this insightful suggestion and have changed the sentence to “and likely circuit wiring” (see line 28 page 1 in the revised paper).

INTRODUCTION, PAGE 2: “Finally, all members of the teneurin family have been associated [...] disorders” should be written as “Finally, (GENETIC ABNORMALITIES OF) all members of the teneurin family have been associated [...] disorders” or something to this effect. It’s not the native protein that cause autism or essential tremors but some abnormality of it.

We have modified the text, and the revised sentence now reads: “Finally, genetic abnormalities of all members of the teneurin family have been associated with diverse neurological disorders.” We also include a citation to teneurin-2-related pathology as requested by reviewer 3 (see lines 48-50 on page 2 of the revised manuscript).

PAGE 4, RESULTS: To examine the structural basis [...] the complete extracellular domain (ECD, Δ 1-341 residues). I suggest to define this construct as ECD, 342-2715. First, it will indicate what it is, rather than what it isn’t, and second, it will give an immediate idea of the size, as I was initially confused by thinking that Ten3 was only 341 residues long.

We have updated the description of the ECD to “ECD, residues 342-2715” (line 124 on page 4 of the revised manuscript) and have also rephrased the number of the monomeric teneurin on pages 10 and 16 of the manuscript to improve clarity.

PAGE 4, RESULTS: The interface is mainly composed of hydrophobic and -phobic surface. What does this (-phobic) mean?

We thank the reviewer for noticing this and the revised sentence now reads: “The interface is mainly composed of hydrophilic and hydrophobic surface.” (see line 136 on page 5 of the updated manuscript)

PAGE 4, RESULTS: numbering is probably wrong the EGF8 domain (residues 762-764). Only 3 residues? ABD domain (residues 2595-2597) Only 3 residues?

This numbering refers to the beta-strands of the respective domain that participate in the formation of the extended beta-sheet: EGF8 domain (residues 761-764); ABD domain (residues 2594-2599). We have clarified this in the manuscript “an extended edge-to-edge β -sheet between residues 761-764 in EGF8 and residues 2594-2599 in the ABD of the two respective subunits” (see lines 138-139 on page 5 of the revised manuscript)

PAGE 5: The main difference [...] of the two SUBUNITS with respect to each other (Fig. 3A)”. I suggest to use the term monomer or protomer instead of SUBUNIT, as in such a large and multidomain protein subunit may lead to some confusion.

We thank the reviewer for the suggestion. However, we prefer to use the word SUBUNIT to describe the individual protein chains. We have now defined this nomenclature in the introduction where we introduce the term SUBUNIT: “subunit refers to one of the two chains in the covalent homodimer” (see lines 68-69 on page 3 of the revised manuscript).

BOTTOM OF PAGE 5: In absence of splice insert A (A0B0 and A1B1 isoforms). Should that be A0B1?

We thank the reviewer for pointing this out. We have updated the sentence as requested (see line 185 on page 6 of the revised manuscript).

In addition to ref 42, this one should be cited as well: Lloyd BA, Han Y, Roth R, Zhang B, Aoto J. Neurexin-3 subsynaptic densities are spatially distinct from Neurexin-1 and essential for excitatory synapse nanoscale organization in the hippocampus. Nat Commun. 2023 Aug 5;14(1):4706. doi: 10.1038/s41467-023-40419-2. PMID: 37543682; PMCID: PMC10404257.

The reference to Lloyd et al. has been added to the manuscript (see reference 52, line 356 on page 12 of the revised manuscript).

Reviewer #2 (Remarks to the Author):

Synapse formation and elimination are intricately orchestrated by an array of cell-adhesion molecules (CAMs), essential in cell-cell interactions and the construction of neuronal networks. Among these CAMs, teneurins are recognized as key recognition molecules, crucial for enabling trans-synaptic communication. Teneurins are characterized by alternative-splicing isoforms, whose unique shapes play a pivotal role in molecular recognition. This influences the binding specificity of partners and plays a key role in synaptic differentiation. Specifically, mouse teneurins-3 has two splice insertion sites: one between EGF7 and EGF8 (splice insert A), and another between the first and second blade of the NHL β -propeller (splice insert B). In this study, Gogou et al. utilized a trio of techniques: Cryo-Electron Microscopy (Cryo-EM), X-ray scattering (SAXS), and negative stain EM, to decipher four structures of mouse teneurin-3 homodimer complexes, each with distinct splice variants combinations. The team first resolved the structure of a compact teneurin-3 A1B1 dimer using Cryo-EM and further investigated the unpacked versions of all four splice isoforms, integrating SAXS and negative stain EM to summarize the dimer models of each splice variant. Subsequent clustering assays assisted them to model higher-order interactions formed by dimers of dimers. Finally, stripe assays revealed that splice insert B alone can guide axon growth. This research offers crucial insights into how short splice inserts modify the conformation of teneurins, altering their dimerization formats and exposing distinct protein surfaces for binding with other proteins. This forms the foundation for specific cellular interactions. Overall, the structures presented in this study are of high quality and provide important insights into how splicing-dependent assembly formations contribute to cell-specific recognition and the development of the neuronal network. While the work is clearly presented and well-executed, there are some major concerns.

Major concerns:

1. While the A1B1 structure of teneurin-3 is derived directly from high-resolution Cryo-EM, the other three dimer models (A1B0, A0B0 and A0B1) are constructed from indirect data obtained through SAXS and negative stain EM. Analysis of Figure 5 suggests that most of the model aligns well with the low-resolution 3D reconstructions, yet some uncertainty remains due to the low resolution of the map. It would be beneficial if the authors could provide additional analysis to validate the accuracy of the proposed dimer assembly formats, thereby minimizing ambiguity stemming from the low resolution.

We would like to thank the reviewer for acknowledging the new insights provided in our paper and the high quality of the data. We understand the concern of the reviewer regarding the validation of the low-resolution maps for variants A0B0, A0B1 and A1B0. We have now added new data that describes the high-resolution cryo-EM map of the compact dimer for splice variant A1B0 (see lines 130-132 on page 5, the corresponding methods section on reference-based particle picking at lines 471-473 on page 18, and Supplemental Figure 2 in the updated manuscript). Just like the A1B1 compact dimer, the new map corresponds well with the isoform's low-resolution map in the previously submitted version. This should further demonstrate the representativity of negative stain EM in identifying compact conformations that are disrupted in the cryo-EM conditions.

In addition, we have determined model-map correlations for each model against all negative stain-derived maps (see table in Supplemental Fig. 6C). The reviewer the correlations quantitatively validate our proposed dimer assembly formats. This is now referred to in the results and methods section (see lines 228-232 on page 8 and lines 549-553 on page 21, respectively, of the revised manuscript).

2. The study identifies four divergent structural assemblies arising from different splice variants of mouse teneurin-3. However, it remains to be demonstrated whether these conformations exist within an animal

and their physiological relevance. The authors are encouraged to provide evidence, such as some mutagenesis or methodologies that could affirm the physiological significance of these conformations.

We thank the reviewer for raising this issue. We would first like to point out that research in the group of Liqun Luo at Stanford University has demonstrated that the different splice variants of Teneurin3 are indeed present in the hippocampus (Berns et al 2018). We apologize for not mentioning this information in the previously submitted manuscript and have added this information in the introduction of the revised manuscript: "Berns and colleagues performed cDNA sequencing of postnatal mouse brain to demonstrate that all teneurin-3 A and B splice variants are expressed in mouse subiculum, and all except A₀B₀ are found in the CA1 region of the hippocampus" (line 88-90 on page 3 of the revised manuscript). We have additionally added: "HEK-neuron co-culturing experiments revealed that overexpression of Ten2-B₀ induces excitatory post-synaptic specialisations in neurons, whereas overexpression of Ten2-A₀B₁ induces inhibitory synapse formation" (see lines 102-104 on page 4 of the updated manuscript) to further emphasize the physiological relevance of teneurin alternative splicing.

Furthermore, we are also very much interested in whether the divergent conformations exist within an animal and towards that end we have been discussing with collaborators to generate isoform-specific antibodies. Unfortunately, generating antibodies for heavily glycosylated proteins, such as Teneurins, is a challenge by itself and for generating isoform-specific antibodies to teneurin-3 splice variants many hurdles still have to be taken. Finally, generation of specific mutations that can (de)stabilize the interface in one isoform but not the other, requires high-resolution structures of all four variants. This has been challenging due to the air-water interface issues (see revised manuscript line 180), but we are keen on tackling these issues in the future. We indeed also envision subsequent generation of animal models with these mutants to further delineate the function of isoforms *in vivo*, but, these are all very long-term prospects, and may serve as complete projects of their own.

3. The paper briefly outlines the binding modes of isoform-specific teneurin-latrophilin trans-cellular complexes. Given that FLRT is also involved in transsynaptic interactions, it would be valuable if the authors could expand on their predictions or descriptions regarding how FLRT interacts with the teneurin-latrophilin complex. Such information would deepen the understanding of these complex molecular interactions.

We thank the reviewer for this suggestion. We have added analysis of the FLRT interaction in an updated Supplemental Fig. 8B of the updated manuscript and discuss it in the corresponding results sections (see lines 274-275 and 280-282 on page 9 of the updated manuscript). In short, our compact conformations would not sterically inhibit FLRT binding to a teneurin-Lphn complex. However, as already described, the Lphn binding site in the Ten3-A₀B₁ form is directed away from the opposing membrane and may thus be unfavorable for Lphn binding, essentially also precluding Lphn-mediated FLRT interaction to Ten3 A₀B₁.

Minor points:

1. Regarding Figure 4B, I suggest adjusting the color representation of EGF8 in the density maps of AOB0 and AOB1 to blue. This change would ensure consistency in the depiction of EGF8 across Figures 4B, 4D, and 4E, making it easier for readers to follow and compare the features across these figures.

We thank the reviewer for this suggestion and have updated the color representation of EGF8 depending on the presence of splice insert A throughout Figure 4 for consistency and clarity.

2. In Figure 6C and Supplementary Figure 5B, the current map format lacks clarity, making it challenging to discern the conformation and interaction details. Specifically, it's difficult to identify the localization of each subunit and the contact points between the two dimers. To improve this, I recommend that the author consider revising them to incorporate either the model alone or a combination of the model with a transparent overlay of the map. It would likely provide a clearer and more detailed visualization of the assembly, facilitating a better understanding of the structures and interactions depicted.

We thank the reviewer for pointing out this unclarity. We have altered the representation of the models and maps by making the maps more transparent, by representing protein structures in 'licorice' representation to remove obscuring structural details unnecessary for overall comparison of subunit orientation (Fig. 5E and Supplemental Fig. 7D of revised manuscript). Additionally, we have coloured the domains as in Figure 1 and labelled the NHL and ABD domains – which are most distinctly sized and organised in the A1Bx conformations - to accentuate the relative reorganisations of the subunits within the isoform-specific conformations. Finally, we also coloured the domain names in the conformation descriptors at the bottom.

Reviewer #3 (Remarks to the Author):

The manuscript entitled “Alternative splicing controls teneurin-3 compact dimer formation for neuronal recognition” by Gogou et al, addresses an important aspect of Neuronal network formation by looking at alternative splicing structural consequences on the teneurin family of cell adhesion molecules, by determining cryoEM structures and biophysical and functional assays with Teneurin3. That way, it offers insights into homo- and heterophilic interactions and thus on neuronal recognition and circuit wiring.

The paper is extremely well-written and is easy to follow. Moreover, the figures are carefully displayed and clearly illustrate the results and conclusions. There are however some aspects that could still be improved, such as:

Lines 48 to 50. If all members of the teneurin family have been associated with neurological disorders, why not name one to teneurin-2? Since all other 3 are illustrated with examples.

We thank the reviewer for the comment and agree that teneurin-2 should also be addressed in the list. We have therefore added a Ten2-related pathology with corresponding citation (see line 50 on page 2 of the revised manuscript, teneurin-2 is associated with major depression).

1) Lines 51 to 58. The author did a good job describing the domain organization of Ten3 but left the intracellular domain (ICD) out. It should be introduced, and if relevant literature exists, it should be cited.

We thank the reviewer for the comment and agree that added discourse on the ICD would be in place for completeness. We therefore added with relevant literature citations : “The teneurin intracellular domain (ICD), approximately 45 kDa in size, exhibits less conservation across different homologues compared to the extracellular domain (ECD). However, common features and domains can be identified, such as an EF-hand-like Ca^{2+} -binding site, potential phosphorylation sites, and polyproline-rich regions known to bind SH3-containing adaptor proteins. The ICD may be released by a mechanism known as regulated intramembrane proteolysis and result in translocation of the ICD to the nucleus where it can coordinate transcription”(see lines 52-57 on page 2 of the revised manuscript).

2) Calcium is used for protein purification and tested in the melting temperature assays and SAXS, but its influence on the structure/function of the protein is not adequately described in the introduction. In my opinion the authors should add a bit of information on this matter.

We thank the reviewer for the suggestion. We have added to the introduction what is known about the effect of calcium on teneurin stability, namely: “Further X-ray crystallography analysis of this teneurin-4 C-rich domain has revealed that three calcium ions are coordinated by eight acidic residues in the C-rich domain. These residues are conserved in mouse and human Teneurin paralogues. SAXS analysis showed that calcium binding stabilizes the compact dimer conformation of teneurin-4.” (see lines 69-72 on page 3 of the revised manuscript)

3) Line 156. It should be A_0B_0 and A_0B_1

We thank this reviewer (and reviewer 1) for their vigilance and have updated the manuscript.

4) Line 175 should be "...and supplemental Fig. 5B"

We have updated the numbering throughout the entire revised manuscript.

5) Figure 6C- In the panel on the right, the 2 transmembrane helices overlap. Is this a mistake or it reflect the position after rotation of the full dimer, and the TM are one in front of the other, from the viewpoint of the reader?

We thank the reviewer for noticing the detail. The helices were added schematically to represent a membrane-embedded molecule, but the TM helices themselves are not model-based. We have separated the helices as not to suggest overlap.

6) Figure 7 A – are both scale bars 400 μm ? Should not the II-Hoechst represent a smaller value?

Yes, indeed. We have added the correct value of 50 μm for high-magnifications I and II.

7) In methods and on the "Stripe assays" is written "Ten3-B0/B1 monomeric (referred to as Ten3 from now on)". If there were many entries after it was understandable, but since it is named only twice after, I think using the full description "Ten3-B0/B1 monomeric" would be better.

We thank the reviewer for the suggestion and have used only the full description. More specifically we have named "monomeric Ten3-B₀/B₁" in lines 608, 609, and 630 on pages 22-23 of the revised manuscript.

Minor things:

We thank the reviewer for their vigilance and have corrected the spelling and nomenclature in all cases mentioned below.

Line 389 (425). A space is missing

Line 426 A_1B_1 , the numbers should be below the line (subscript). It happens twice in the same line.

Line 436 A_1B_1 (see above)

Line 483 a space too much (20- mM)

Line 517 & 575 CO₂ – the 2 should be subscript

Line 549 MgCl₂ – the 2 should be subscript

Additional modifications

- We have improved the resolution of the A1B1 compact dimer cryo-EM reconstruction by introducing a round of 3D reference-based particle picking to increase the number of particles in the reconstruction (see Supplemental Fig. 1D and methods in the revised manuscript). The atomic model was updated accordingly and resulted in small differences in interface residues identified (see results Fig. 1, Fig. 2, Fig. 3, and Supplemental Fig. 3B in revised manuscript) and the text in the corresponding results section.
- Style and angles of the interface insets of atomic model in Fig. 2 of revised manuscript have been altered for clarity.
- The residue numbering to indicate the isoform length at the end of the linear representations of Fig. 4A have been updated according to isoform numbering conventions.
- Small changes were made in the methods and figure legends describing the stripe assays, and the references from the methods section have also been added to the stripe assays results section.
- Supplemental Fig. 6 was moved up (previously Supplemental Fig. 8) to match the storyline in the manuscript.

REVIEWERS' COMMENTS

Reviewer #1 (Remarks to the Author):

I'd like to thank the authors for a complete and thorough revision of this manuscript. I don't have any other concern and I recommend publication as is.

Reviewer #2 (Remarks to the Author):

the author did a good job and addressed all my concerns. At this time, I recommend publication.

Reviewer #3 (Remarks to the Author):

In their revision, the authors have adequately addressed my comments.